# Prediction error and repetition suppression have distinct effects on neural representations of visual information

Matthew F Tang[1,2]*, Cooper A Smout[1,2], Ehsan Arabzadeh[2,3], Jason B Mattingley[1,2,4]

[1]Queensland Brain Institute, The University of Queensland, St Lucia, Australia; [2]Australian Research Council Centre of Excellence for Integrative Brain Function, Victoria, Australia; [3]Eccles Institute of Neuroscience, John Curtin School of Medical Research, The Australian National University, Canberra, Australia; [4]School of Psychology, The University of Queensland, St Lucia, Australia

**Abstract** Predictive coding theories argue that recent experience establishes expectations in the brain that generate *prediction errors* when violated. Prediction errors provide a possible explanation for *repetition suppression*, where evoked neural activity is attenuated across repeated presentations of the same stimulus. The predictive coding account argues repetition suppression arises because repeated stimuli are expected, whereas non-repeated stimuli are unexpected and thus elicit larger neural responses. Here, we employed electroencephalography in humans to test the predictive coding account of repetition suppression by presenting sequences of visual gratings with orientations that were expected either to repeat or change in separate blocks of trials. We applied multivariate forward modelling to determine how orientation selectivity was affected by repetition and prediction. Unexpected stimuli were associated with significantly enhanced orientation selectivity, whereas selectivity was unaffected for repeated stimuli. Our results suggest that repetition suppression and expectation have separable effects on neural representations of visual feature information.
DOI: https://doi.org/10.7554/eLife.33123.001

*For correspondence:
m.tang1@uq.edu.au

## Introduction

At any moment in time, the brain receives more sensory information than can be responded to, creating the need for selection and efficient processing of incoming signals. One mechanism by which the brain might reduce its information processing load is to encode successive presentations of the same stimulus in a more efficient form, a process known as *neural adaptation* (*Fairhall et al., 2001*; *Kvale and Schreiner, 2004*; *Smirnakis et al., 1997*). Such adaptation has been observed across different sensory modalities and species, and has been suggested as a potential mechanism for enhancing the coding efficiency of individual neurons and neuronal populations (*Adibi et al., 2013*; *Benucci et al., 2013*; *Maravall et al., 2007*). A particular form of neuronal adaptation, known as *repetition suppression*, is characterised by attenuated neural responses to repeated presentations of the same stimulus (*Diederen et al., 2016*; *Gross et al., 1967*; *Keller et al., 2017*; *Movshon and Lennie, 1979*; *Rasmussen et al., 2017*). Here, we asked whether predictive coding theory, which assumes that sensory processing is influenced by prior exposure, can account for changes in neural representations observed with repetition suppression.

The phenomenon of repetition suppression has been widely exploited to investigate neural representations of sensory information. Repeated exposures allow for more efficient representation of subsequent stimuli, as manifested in improved behavioural performance despite a significant reduction in neural activity (*Henson and Rugg, 2003*; *Schacter and Buckner, 1998*). Repetition suppression paradigms have been used extensively in human neuroimaging because they are commonly considered to be analogous to the single-cell adaptation effects observed in animal models (see *Barron et al., 2016* for review). The exact relationship between the effects seen in human neuroimaging studies and animal neurophysiology has, however, yet to be fully established.

The view that repetition suppression observed in human neuroimaging studies reflects neuronal adaptation has recently been challenged by hierarchical predictive coding theories (*Auksztulewicz and Friston, 2016*; *Summerfield et al., 2008*). These theories argue that the brain interprets incoming sensory events based on what would be expected from the recent history of exposure to such stimuli (*Friston, 2005*; *Rao and Ballard, 1999*). According to these theories, predictions are generated within each cortical area and are bi-directionally propagated between higher and lower areas, including to primary sensory regions, allowing for more efficient representation of expected stimuli. When there is a precise expectation, incoming information can be efficiently represented by recruiting a small pool of relevant neurons (*Friston, 2005*). When there is a mismatch between an expectation and the stimulus presented, that is, when there is a *prediction error*, the stimulus is less efficiently represented and thus elicits a larger neural response.

The majority of evidence for predictive coding comes from human neuroimaging experiments in which the presentation of an unexpected stimulus generates a larger response than the presentation of an expected stimulus. In studies employing electroencephalography (EEG) and magnetoencephalography (MEG), this effect is known as the *mismatch negativity* (*Garrido et al., 2009*; *Näätänen et al., 2007*; *Wacongne et al., 2011*), where an unexpected stimulus evokes significantly greater negativity than an expected stimulus. To date, however, no study has tested a key premise of predictive coding, namely, that expected stimuli are more efficiently encoded in the brain relative to unexpected stimuli, in terms of their elementary feature representations. Nor has any previous investigation examined whether the mismatch negativity response is associated with a change in neural tuning to stimulus features such as orientation.

To test the hypothesis that prediction error can account for repetition suppression effects, *Summerfield et al. (2008)* introduced an experimental paradigm in which the identity of a face stimulus was either repeated in 80% of trials (making the repetition *expected*) or was changed in 80% of trials (making the repetition *unexpected*). There was greater attenuation of the BOLD response in the fusiform face area when a face repetition was expected, relative to when it was unexpected, suggesting that repetition suppression is reduced by unexpected stimuli. This attenuation of repetition suppression by failures of expectation has also been replicated using fMRI (*Larsson and Smith, 2012*) and M/EEG, using high-level stimuli such as faces (*Summerfield et al., 2011*), and simple stimuli such as auditory tones (*Todorovic and de Lange, 2012*; *Todorovic et al., 2011*).

A potential reconciliation of the relationship between expectation and repetition suppression comes from work showing that while expectations decrease the overall amount of neural activity, they can also yield sharper representations of sensory stimuli (*Kok et al., 2012*). This work goes beyond conventional neuroimaging approaches, which typically only measure overall levels of neural activity (*Buckner et al., 1998*; *Kourtzi and Kanwisher, 2001*; *Tootell et al., 1995*). Such amplitude changes could in principle be produced by one or more different types of change in the underlying neural representations. For instance, both sharpening, where the response to only unpredicted features is suppressed, and gain reduction, where a multiplicative suppression occurs for all features, could be associated with decreased population activity, even though the amount of information carried by the representations will be markedly different. Recently introduced multivariate pattern analytic approaches to human neuroimaging – specifically forward encoding modelling – allow for the quantification of stimulus-selective information contained within patterns of neural activity in human observers (*Brouwer and Heeger, 2009*; *Garcia et al., 2013*; *King et al., 2016*; *Kok et al., 2017*; *Myers et al., 2015*; *Salti et al., 2015*; *Wolff et al., 2017*). This approach goes beyond typical multivariate pattern analyses (which normally produce only accuracy scores) by quantifying neural representations evoked by sensory stimuli to reveal both the accuracy and the *tuning fidelity* for the specific feature-dimension of interest.

Here, we used multivariate forward encoding methods to test whether repetition suppression and expectation have different effects on the way the brain represents visual information, in this case the orientation of grating stimuli. To anticipate the results, we found that soon after stimulus onset, repetition suppression had no effect on visual orientation selectivity, but violated expectations were associated with a significantly increased orientation-selective response through gain modulation, with no corresponding change in response fidelity. This representation was transiently re-activated at around 200 ms post-stimulus onset, suggesting that feedback influences initial sensory encoding of an unexpected stimulus, which in turn allows for updating of the sensory prior.

## Results

We used a modified version of the paradigm introduced by *Summerfield et al. (2008)*, replacing the face stimuli used in that study with oriented Gabors. These low-level stimuli allowed us to quantify the degree of orientation selectivity in EEG activity to determine how the representation of orientation is affected by prediction error and repetition suppression. Each of 15 observers participated in two EEG sessions. On each trial, two Gabors were presented sequentially (100 ms presentation, 600 ms stimulus onset asynchrony), and these stimulus pairs either repeated or alternated in their orientation (*Figure 1A*, *Video 1*). The predictability of the repeated and alternating pairs was varied in a block-wise manner to manipulate expectation. In a *repeating* block, the orientations of the two Gabors in a pair repeated in 80% of trials, and alternated in the remaining 20%. These contingencies were reversed in the *alternating* block (*Figure 1B*). The orientations of successive stimuli across a block were randomised to limit any accumulated effects of adaptation and prediction. As repetition suppression and expectation form orthogonal dimensions of the task, the design allowed us to isolate their respective contributions to neural responses. Participants completed an unrelated task of discriminating (red vs green) rare coloured Gabors (which occurred on 10% of trials).

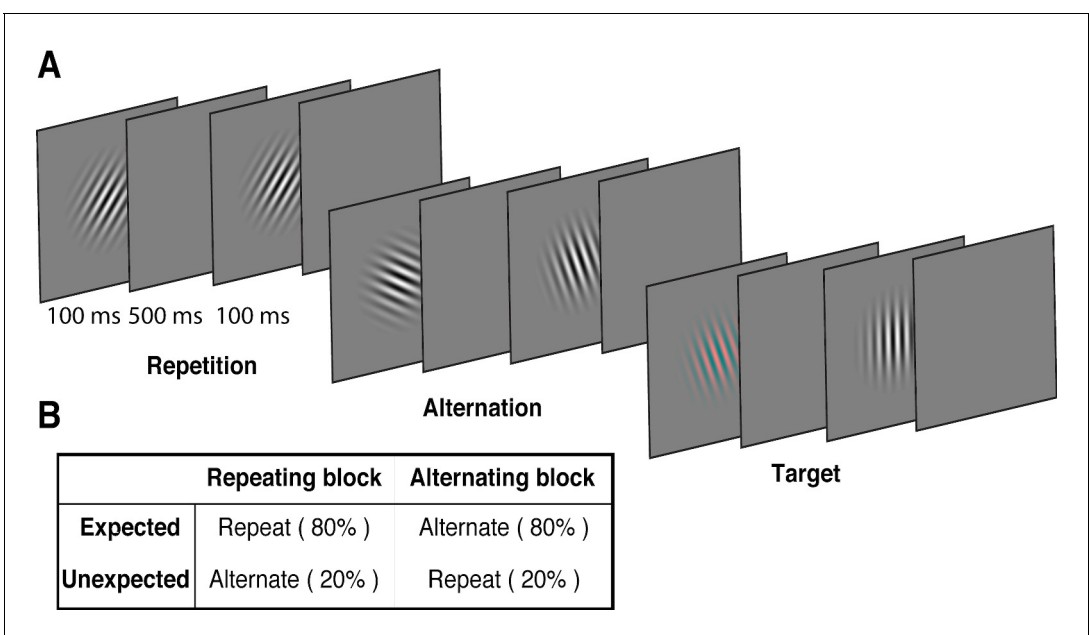

**Figure 1.** Example stimulus displays and task design. (**A**) Schematic of the stimuli and timing used in the experiment. Participants viewed a rapid stream of pairs of Gabors and monitored for an infrequent coloured target (10% of trials). The stimulus orientations were pseudorandomly varied across trials between 0° and 160° (in 20° steps), allowing estimation of orientation-selective information contained within patterns of EEG activity. (**B**) The orientation of the pairs of Gabors could either repeat or alternate. In one type of block, 80% of trials were orientation repeats and the remaining 20% alternated (Repeating blocks); in the other type of block, these contingencies were reversed (Alternating blocks).
DOI: https://doi.org/10.7554/eLife.33123.002

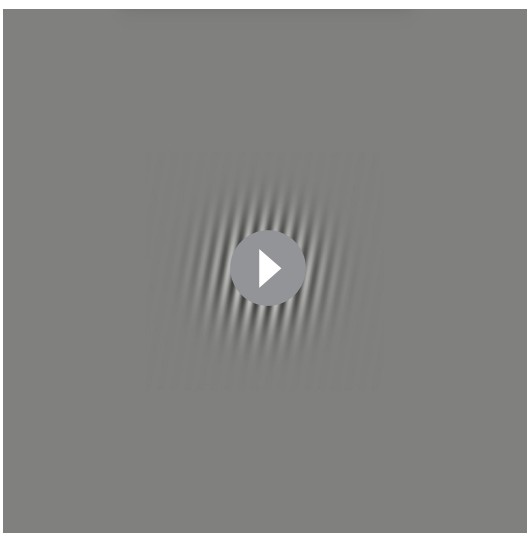

**Video 1.** Example of a stimulus sequence of Gabors in a typical alternating block.
DOI: https://doi.org/10.7554/eLife.33123.003

## Repetition suppression and prediction error affect the overall level of neural activity

The Gabors elicited a large response over occipital-parietal areas (*Figure 2A*). Consistent with previous work (*Cui et al., 2016*; *Keller et al., 2017*; *Rentzeperis et al., 2012*; *Summerfield et al., 2011*; *Todorovic et al., 2011*; *Todorovic and de Lange, 2012*; *Tootell et al., 1998*), there was a significant repetition suppression effect (Repeat < Alternating), such that the response to repeated stimuli was significantly reduced compared with the response to alternating stimuli (*Figure 2A*). The repetition suppression effect was evident over a large cluster of occipital-parietal electrodes at two time intervals: an early effect from 79 to 230 ms, and a later effect from 250 to 540 ms after the onset of the second stimulus (cluster p < 0.025; *Figure 2B* and caption). A large cluster of frontal electrodes mirrored the repetition suppression effect with a similar time course: the ERP over these frontal sites had the same pattern, but was reversed in sign, suggesting it originated from the same dipole as the occipital response.

Also consistent with previous results (*Garrido et al., 2009*; *Summerfield et al., 2011*; *Todorovic et al., 2011*; *Todorovic and de Lange, 2012*), there was a significant expectation effect (Expected < Unexpected). Specifically, there was a significantly greater negativity for unexpected versus expected stimuli, and this effect was most prominent over a cluster of occipital-parietal electrodes around 75–150 ms after stimulus presentation (*Figure 2C*). As with the repetition suppression result described above, there was an expectation effect of opposite polarity over occipital-parietal electrodes. This effect was significant at an early time point post-stimulus (79–130 ms), but not at later time points (320–390 ms; *Figure 2D*). Finally, there was no interaction between repetition suppression and expectation (i.e., no significant positive or negative clusters, all p > 0.05). Taken together, these results reveal both repetition suppression and expectation effects in the neural data, which were indexed separately as shown in *Figure 2*.

We conducted a further traditional peak analysis, to aid comparison with previously published studies on the mismatch negativity (*Garrido et al., 2013*; *Näätänen et al., 2007*; *Saarinen et al., 1992*). We bandpass filtered the ERPs (2–40 Hz) to recover the stereotypic waveform (*Figure 2C*) and examined two classic early components – the N1 and P1 – averaged across a broad grouping of occipital-parietal electrodes (O1, O2, Oz, POz, PO7, PO3, PO8, PO4, P3, Pz, P2). As in previous studies (*Dehaene et al., 2001*; *Caharel et al., 2009*), we defined the P1 as the largest positivity between 80 and 110 ms after stimulus presentation, and the N1 as the largest negativity between 90 and 130 ms after stimulus presentation. A relatively wide temporal window was used to capture inter-individual response variation. As expected, for the P1 component, the repeated stimulus evoked a significantly smaller positivity ($t(14) = 3.03$, $p = 0.009$) than the alternating stimulus (*Figure 2D*), reflecting a repetition suppression effect. There was no such effect of expectation on the P1 ($t(14) = 0.26$, $p = 0.80$). By contrast, as predicted from previous work (*Garrido et al., 2013*; *Näätänen et al., 2007*; *Saarinen et al., 1992*), analysis of the N1 component showed that the unexpected stimulus evoked a significantly greater negativity than the expected stimulus, ($t(14) = 5.75$, $p < 0.0001$). The repetition suppression effect was also present in the N1 ($t(14) = 2.39$, $p = 0.03$), but critically in the opposite direction to that of the expectation effect.

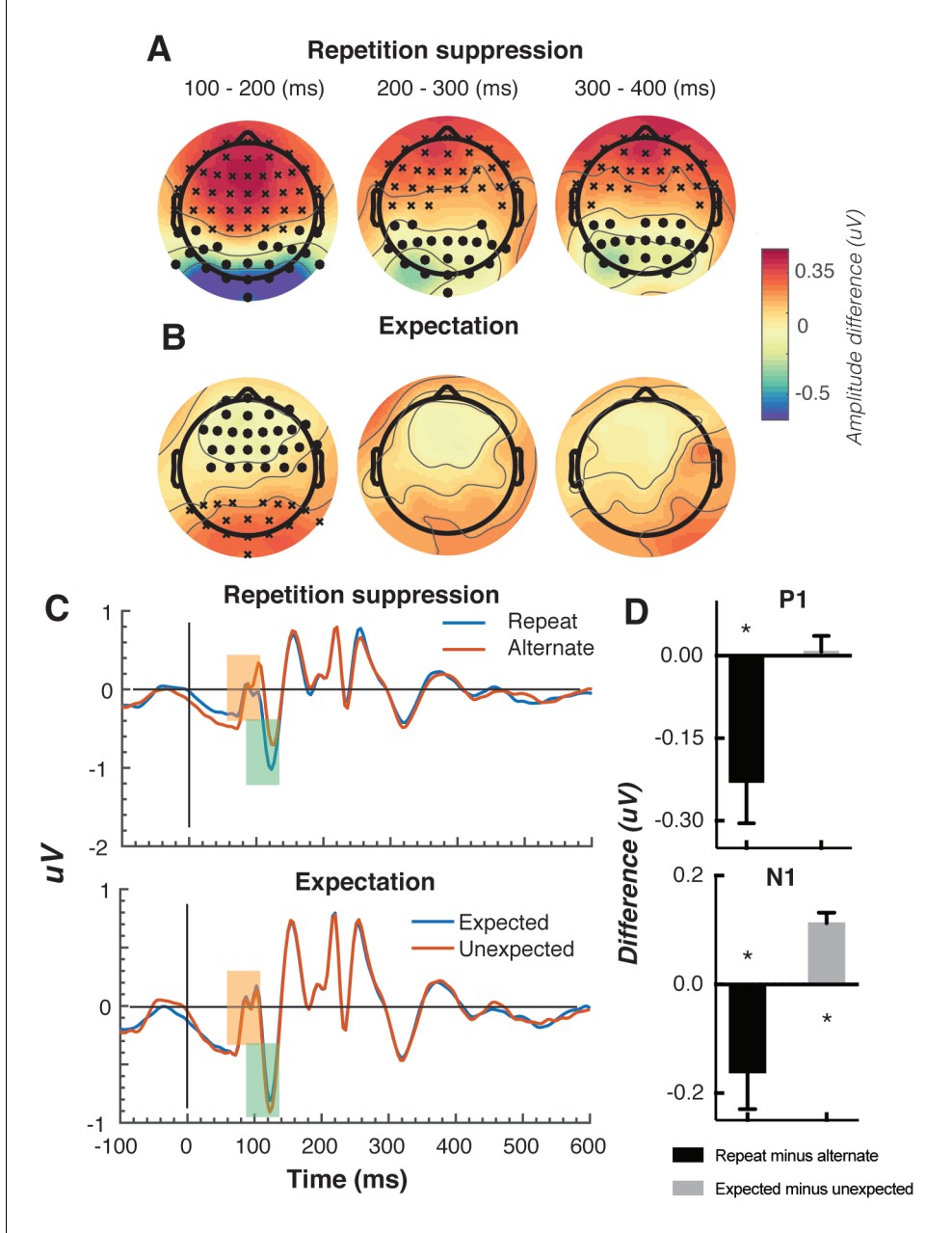

**Figure 2.** Univariate EEG results for the effect of repetition suppression and expectation on the second stimulus in a pair. Panels **A and B** show the main effects of repetition suppression and expectation, respectively, over three post-stimulus epochs (100–200 ms, 200–300 ms, 300–400 ms) and across all electrodes. The main effect of repetition suppression is displayed as Repeating minus Alternating trials. The main effect of expectation is displayed as Expected minus Unexpected trials. Circles indicate clusters of electrodes with significantly reduced activity, and crosses indicate clusters of electrodes with significantly increased activity (alpha $p < 0.05$, cluster $p < 0.025$, N permutations = 1500). (**C**) Bandpass filtered (2–40 Hz) event-related potentials (ERPs) for the two conditions, averaged over occipital-parietal electrodes (O1, O2, Oz, POz, PO7, PO3, PO8, PO4, P3, Pz, P2). A peak analysis was conducted to aid comparison with previous studies. Orange shading indicates the P1 component; green shading indicates the N1 component. (**D**) Peak analysis results for P1 and N1 components. Note that the plotted values represent *differences* between conditions, as indicated, rather than condition-specific evoked responses. Asterisks indicate $p < 0.05$. Error bars indicate ±1 standard error.

DOI: https://doi.org/10.7554/eLife.33123.004

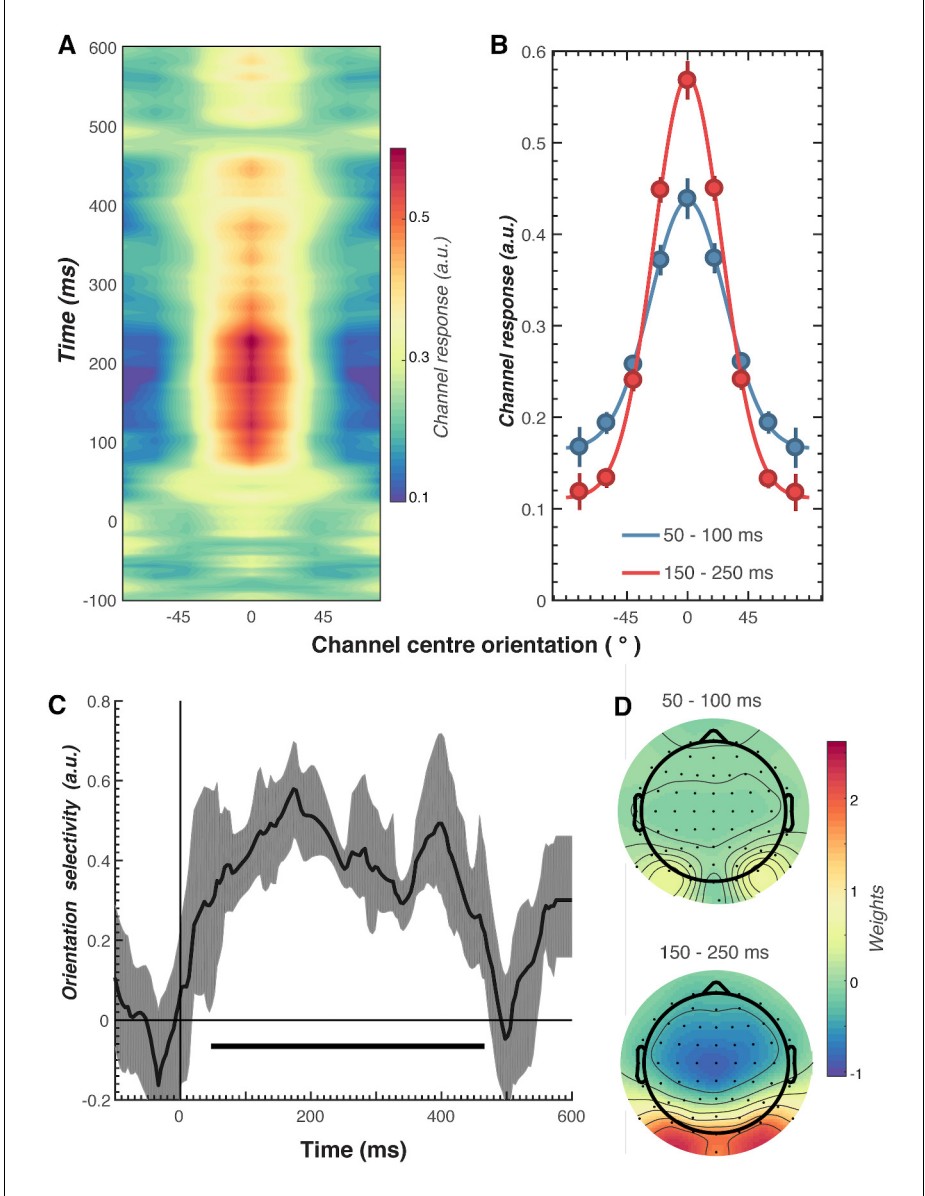

**Figure 3.** Results of the forward encoding modelling for orientation-selectivity. (**A**) Time-resolved orientation tuning curve across all participants and conditions in response to the second Gabor in the pair. The forward encoding approach resulted in a tuning curve for each of the nine presented orientations. These tuning curves were then centred at each presented orientation (here labelled as 0°) to combine across all orientations. The orientation-selective response is contained within the overall pattern of EEG; activity begins soon after stimulus onset and peaks at around 250 ms before declining. (**B**) Population tuning curve of the stimulus reconstruction across participants, averaged between 50–100 ms and 150–250 ms after stimulus presentation. Each line is a fitted Gaussian response with a variable offset used to quantify orientation selectivity. Error bars indicate ±1 standard error of the mean across participants. (**C**) Amplitude of the channel response over time, averaged across all conditions (black line). The thick black line indicates significant encoding of stimulus orientation based on a cluster-permutation test across participants (cluster p < 0.05, N permutations = 20,000). Encoding accuracy was reliable from 52 to 470 ms post-stimulus onset. The error shading (in grey) indicates bootstrapped 95% confidence intervals of the mean. (**D**) Topographic plots of the weights (averaged across the nine orientation channels across all participants) derived from forward encoding at the corresponding time points shown in panel B. (a.u. = arbitrary units).

DOI: https://doi.org/10.7554/eLife.33123.005

## Expectations increase orientation-selective information contained within patterns of EEG activity

We next examined the key question of whether repetition suppression and expectation differentially affect neural representations of orientation information. To do this, we used a forward encoding

approach to reconstruct orientation-selective information contained within the multivariate pattern of EEG activity distributed across the scalp (*Figure 3*; see Materials and methods for details). Briefly, this technique transforms EEG sensor-level responses into tuned 'feature' channels (*Brouwer and Heeger, 2009*; *Garcia et al., 2013*; *Kay et al., 2008*; *Myers et al., 2015*), in this case, orientation-selective features. For each trial, the presented orientation was convolved with a canonical, orientation-selective tuning function and regressed against the pattern of EEG activity across all sensors at each time point. This created a spatial filter of the multivariate EEG activity that differentiated orientations (*Figure 3D*). These weights were then inverted to reconstruct the model and multiplied against an independent set of test trials to produce responses in the modelled orientation channels. These sets of responses were then used to evaluate the degree of orientation selectivity in those trials. The procedure was repeated for all time points in the trial, and a cross-validated approach was used until all trials had been used for both training and testing.

As shown in *Figure 3*, the forward encoding revealed a strong, orientation-selective response derived from the multivariate pattern of EEG activity. This orientation-tuned response was evident from ~50 to ~470 ms after stimulus onset, and peaked between ~120 and 250 ms (*Figure 3C*). Examination of the regression weights revealed that this response was largely driven by activity centred over occipital-parietal areas (*Figure 3D*).

To examine our central question of whether repetition suppression and expectation have differential effects on neural representations of orientation, we split and averaged the results of the forward encoding by trial type, and fitted these with Gaussians (see Materials and methods) to quantify orientation selectivity (*Figure 4*). Repetition suppression did not affect the amount of orientation selectivity contained within the EEG data, with similar selectivity for repeated and alternating trials. This was the case even though the repeated trials had a markedly smaller EEG response over occipital and parietal electrodes (see *Figure 2A*), where the forward encoding model was maximally sensitive. This result is consistent with the 'efficient representation' hypothesis of repetition suppression (*Gotts et al., 2012*), which argues that the overall neural response is smaller with repetition suppression due to more efficient coding of stimulus information.

Examining the effect of expectation revealed a markedly different pattern of results. As shown in *Figure 4A*, at 79–185 ms after the onset of the second stimulus in the pair, orientation-selectivity increased significantly (p < 0.0001) when the stimulus was unexpected relative to when it was expected, and this effect arose at the earliest stages of the brain's response to that stimulus. Moreover, the expectation signal contained enhanced information about the specific features of the stimulus that violated the expectation, in this case the orientation of the second grating. We conducted the same statistical tests on the three other parameters defining the Gaussian function (namely, the width, centre orientation and baseline) to determine how repetition suppression and expectation might affect other properties of the neural representation. There was no reliable influence of repetition suppression on any of these Gaussian parameters (all ps > 0.32). For expectation, there was a significant decrease in baseline activity over the same time period as observed for the increase in amplitude (79–185 ms, p = 0.001), but there were no significant effects for the other parameters (all ps > 0.30).

We followed up this initial analysis to ensure we did not miss any small effects of repetition suppression or expectation on any aspects of stimulus representation. We increased the signal-to-noise by averaging the stimulus reconstruction over this early time period (79–185 ms after stimulus presentation), and fitted Gaussians to each participant's data individually (*Figure 4B*). This again showed that the amplitude of the response was significantly ($t(14)$ = 3.34, p = 0.005) higher for unexpected ($M$ = 0.67, $SE$ = 0.06) than for expected ($M$ = 0.41, $SE$ = 0.03) stimuli. By contrast, the width of the representations was similar for unexpected ($M$ = 29.62°, $SE$ = 4.72°) and expected ($M$ = 26.72°, $SE$ = 2.74°) stimuli, ($t(14)$ = 0.78, p = 0.45). There was also a small but non-significant ($t(14)$ = 1.94, p = 0.06) trend for a smaller baseline response (i.e., non-orientation tuned activity) in the unexpected ($M$ = −0.01, $SE$ = 0.07) than in the expected ($M$ = 0.13, $SE$ = 0.02) condition. For comparison, we also averaged the same time period for the repetition suppression conditions, and found similar curves for the repeated and alternating trials (*all* ps > 0.18). This analysis confirms the previous result, which employed more conservative nonparametric cluster-based testing.

It might be argued that the particular baseline period we chose for the encoding analyses - namely from −100 to 0 ms before the onset of the second Gabor in each pair – biased the results by incorporating a purely top-down expectation template triggered by the orientation of the first

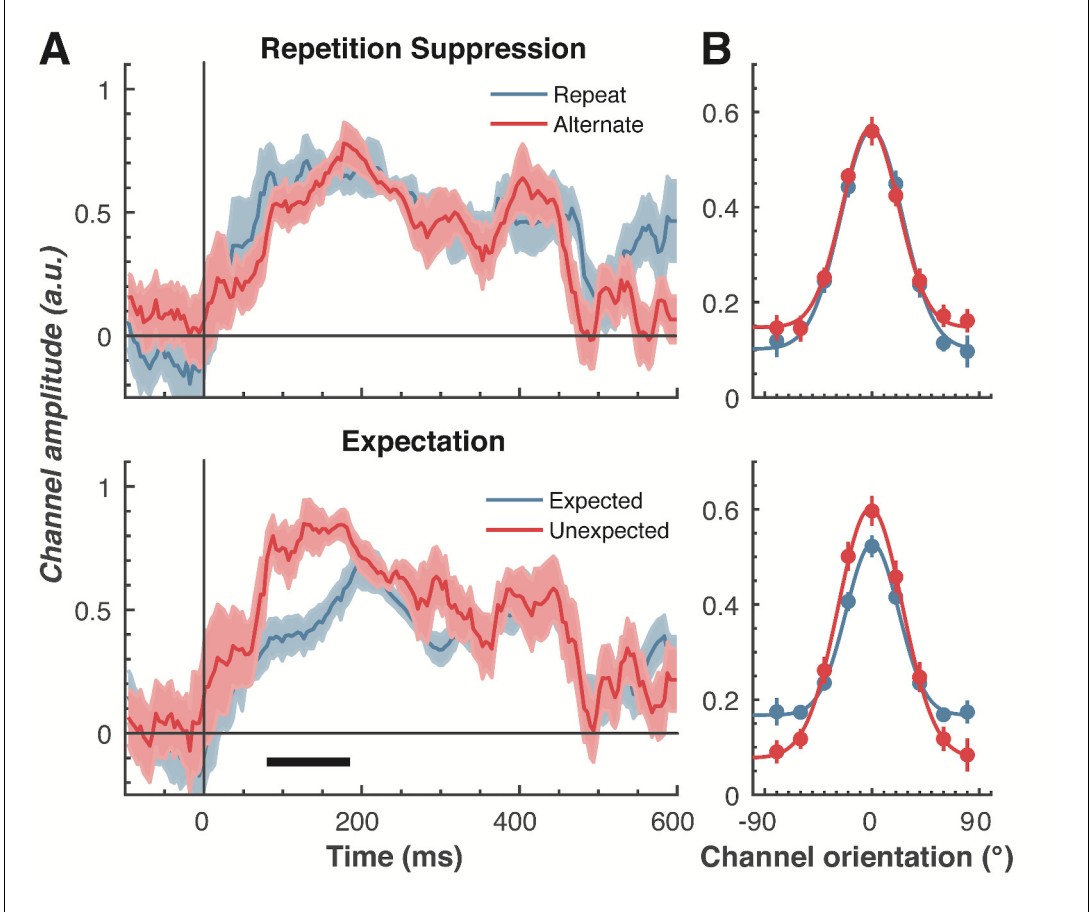

**Figure 4.** The effect of repetition suppression and expectation on orientation selectivity measured using forward encoding modelling. (**A**) Amount of orientation-selective information (given by the amplitude of the fitted Gaussian) from the EEG signal in response to the second Gabor in a pair, shown separately for repetition suppression (upper panel) and expectation (lower panel). The thick black line indicates significant differences between the conditions (two-tailed cluster-permutation, alpha $p < 0.05$, cluster alpha $p < 0.05$, N permutations = 20,000). (**B**) Population tuning curves averaged over the significant time period (79–185 ms) depicted in panel A. The curves, shown as fitted Gaussians, illustrate how overall stimulus representations are affected by repetition and expectation. While there was no difference in orientation tuning for repeated versus alternate stimuli (upper panel), the amplitude of the orientation response increased significantly, and the baseline decreased, for unexpected relative to expected stimuli. Error bars indicate ±1 standard error.

DOI: https://doi.org/10.7554/eLife.33123.006

The following figure supplement is available for figure 4:

**Figure supplement 1.** The effect of a different baseline period (−100 to 0 ms before onset of the first Gabor) on orientation selectivity for the two main conditions.

DOI: https://doi.org/10.7554/eLife.33123.007

Gabor (**Kok et al., 2017**). To rule out this possibility, we performed a further forward encoding analysis where we baselined the raw EEG data to the mean activity from −100 to 0 ms before the *first* Gabor in each pair. Critically, this control analysis involved a baseline period over which it was not possible to form a top-down expectation of the orientation of the second Gabor based on the orientation of the first. This analysis yielded the same pattern of results as the original analysis (*Figure 4— figure supplement 1*), such that the unexpected stimulus evoked significantly greater orientation selectivity than the expected stimulus ($p = 0.02$). Also in line with the original analyses, the width of the representation was not affected by expectation ($p = 0.44$), and there was no effect of repetition suppression on orientation selectivity ($p = 0.64$). We can thus be confident that the effect of expectation on orientation selectivity that we report here, based on our forward encoding analyses, is not an artefact of the baselining procedure.

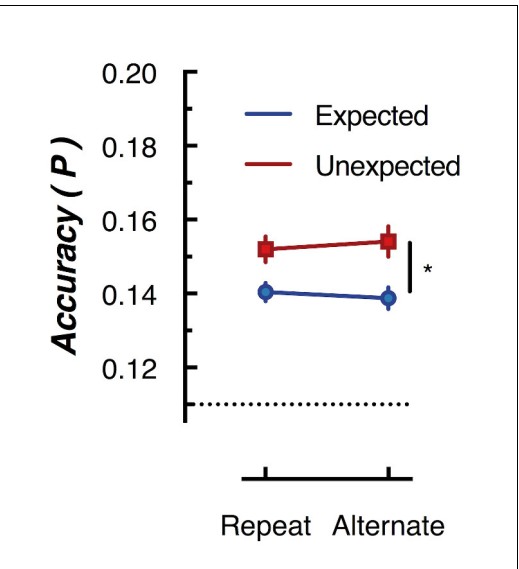

**Figure 5.** Peak (naive Bayes) classification accuracy of the presented grating orientation for expected and unexpected conditions. The dotted line indicates chance performance (1/9 orientations). The error bars indicate ±1 standard error of the mean.

DOI: https://doi.org/10.7554/eLife.33123.008

We also used a number of approaches to determine whether repetition suppression and expectation interacted to affect orientation selectivity. First, we took the difference scores between the combination of factors (e.g., expected repetition minus unexpected repetition, and expected alternation minus unexpected alternation) and compared these using the same cluster-based permutation testing outlined above. This analysis revealed no significant interactions between the factors for any parameter (all p$s$ > 0.10). Second, we found the largest orientation-selectivity, defined by the amplitude of the fitted Gaussian, across the 600 ms following stimulus presentation. For each participant, this resulted in a single value for the four conditions. Each of these values was subjected to a two-way repeated-measures ANOVA, which again revealed no significant interaction between the factors (all p$s$ > 0.30)

To further examine whether orientation-selectivity contained within the overall pattern of EEG activity differed for unexpected and expected stimuli, we used multivariate discriminant analysis to perform traditional backward decoding (*Grootswagers et al., 2017*; *Kamitani and Tong, 2005*; *King and Dehaene, 2014*; *Marti et al., 2015*). This approach (*Figure 5*) yielded the same pattern of results as that revealed by the forward encoding approach described above. The same cross-validation procedure was used as in the forward encoding analysis, but accuracy was now defined as the proportion of trials labelled with the correct orientation. To facilitate comparison with the results of *Kok et al. (2012)*, we took the peak classification accuracy within a 600 ms window after presentation of the second grating within each pair. This analysis confirmed the results of the forward encoding: orientations shown in unexpected trials were classified better than orientations shown in expected trials ($F_{(1,14)}$ 76.42, p < 0.00001). Again, there was no effect of repetition on classification accuracy ($F_{(1,14)}$ = 0.027, p = 0.87); nor was there a significant interaction ($F_{(1,14)}$ = 2.52, p = 0.13). This suggests the finding is not specific to the analysis method but rather reflects how expectation affects the representation of sensory information in general.

## Expectation affects the temporal stability of stimulus representations

Next, we examined whether repetition suppression and expectation affected dynamic, ongoing stimulus representations by using cross-temporal generalisation (*King and Dehaene, 2014*; *King et al., 2014*; *Myers et al., 2015*; *Spaak et al., 2017*; *Stokes et al., 2013*). To do this, we used the same forward encoding approach as in the previous analysis, but now the weights were derived from one time point on one set of trials, and then applied at every time point in the test trials. Again, a cross-

validation approach was used, with all trials serving as both training and test. This analysis examined whether the same spatial pattern of EEG activity that allowed for orientation selectivity generalised to other time points, thus revealing whether the pattern of orientation-selective activity was stable or changed over time.

As shown in *Figure 6*, optimal orientation selectivity was on-axis (training time equals test time) between 100 ms and 300 ms after stimulus presentation, suggesting that the stimulus representation changed dynamically over time (*King and Dehaene, 2014*). There was also significant off-axis orientation-selectivity from 100 to 500 ms after stimulus presentation, suggesting that some aspects of the neural representation of orientation were stable over time.

There was no effect of repetition suppression on temporal generalisation of orientation information (upper panels of *Figure 6*), suggesting that repetition suppression did not affect the temporal stability of neural representations of orientation. Examining the effect of expectation on cross-temporal generalisation confirmed that there was significantly more on-axis orientation selectivity when the stimulus was unexpected than when it was expected (cluster p = 0.02). This increased on-axis orientation selectivity generalised off-axis at around 300–400 ms after stimulus onset (cluster p = 0.01), suggesting that the same representation that is activated to process the expectation is reactivated later as the stimulus continues to be processed. Such a signal could constitute the prior of the prediction, as this should be updated on the basis of incoming sensory evidence, which in turn would likely require reactivation of the unexpected stimulus.

## Discussion

Our findings demonstrate that repetition suppression and expectation have distinct effects on neural representations of simple visual stimuli. Repetition suppression had no effect on orientation selectivity, even though the neural response to repeated stimuli was significantly reduced over occipito-parietal areas. Unexpected stimuli, on the other hand, showed significantly increased orientation-selectivity relative to expected stimuli. This same early representation of the unexpected stimulus appeared to be reactivated at 200–300 ms after the initial neural response, supporting the idea that sensory expectations may be updated through comparison with incoming sensory evidence. These results suggest that repetition suppression and expectation are separable and independent neural computations.

Our work provides a significant advance in understanding how predictions allow the brain to process incoming sensory information by comparing what is expected with what actually occurs. How expectations affect neural responses has been extensively investigated using mismatch negativity paradigms in which an unexpected stimulus causes a larger neural response than an expected stimulus (*Bekinschtein et al., 2009*; *Garrido et al., 2009*; *Näätänen et al., 2007*). Such mismatch responses to an unexpected stimulus have often been attributed to the generation of a prediction error that updates expectation based on a conflict between sensory evidence and the prior (*Garrido et al., 2009*). To date, however, most studies have focused exclusively on the overall magnitude of neural responses to unexpected events, rather than assessing the quality of stimulus-specific information potentially contained within such responses. As noted above, enhanced neural activity to unexpected visual events could reflect a differential response to one of a number of possible stimulus features, or simply an increase in baseline activity associated with a non-selective response. By examining how expectation affects the representation of an elementary feature dimension – in this case, orientation – our results imply the operation of at least two distinct neural processes at separate times following stimulus onset. Incoming sensory information is first evaluated against the prior (which occurs early after stimulus presentation). When an unexpected stimulus is detected and generates a prediction error, the representation is amplified through gain enhancement. Later, around 300 ms after stimulus presentation, this same representation is reactivated to update the expectation against the initially predicted representation.

According to predictive coding theory, expected stimuli should be more efficiently represented than unexpected stimuli largely because the reduced neural response still encodes stimuli with the same fidelity (*Friston, 2005*). A more efficient response could be due to sharpening of neuronal tuning to stimulus features, or to a reduction in the gain of evoked neural responses. Our results support the latter interpretation. Specifically, there was no evidence that a fulfilled expectation leads to a sharper representation of orientation information. Our findings might imply that the brain needs to

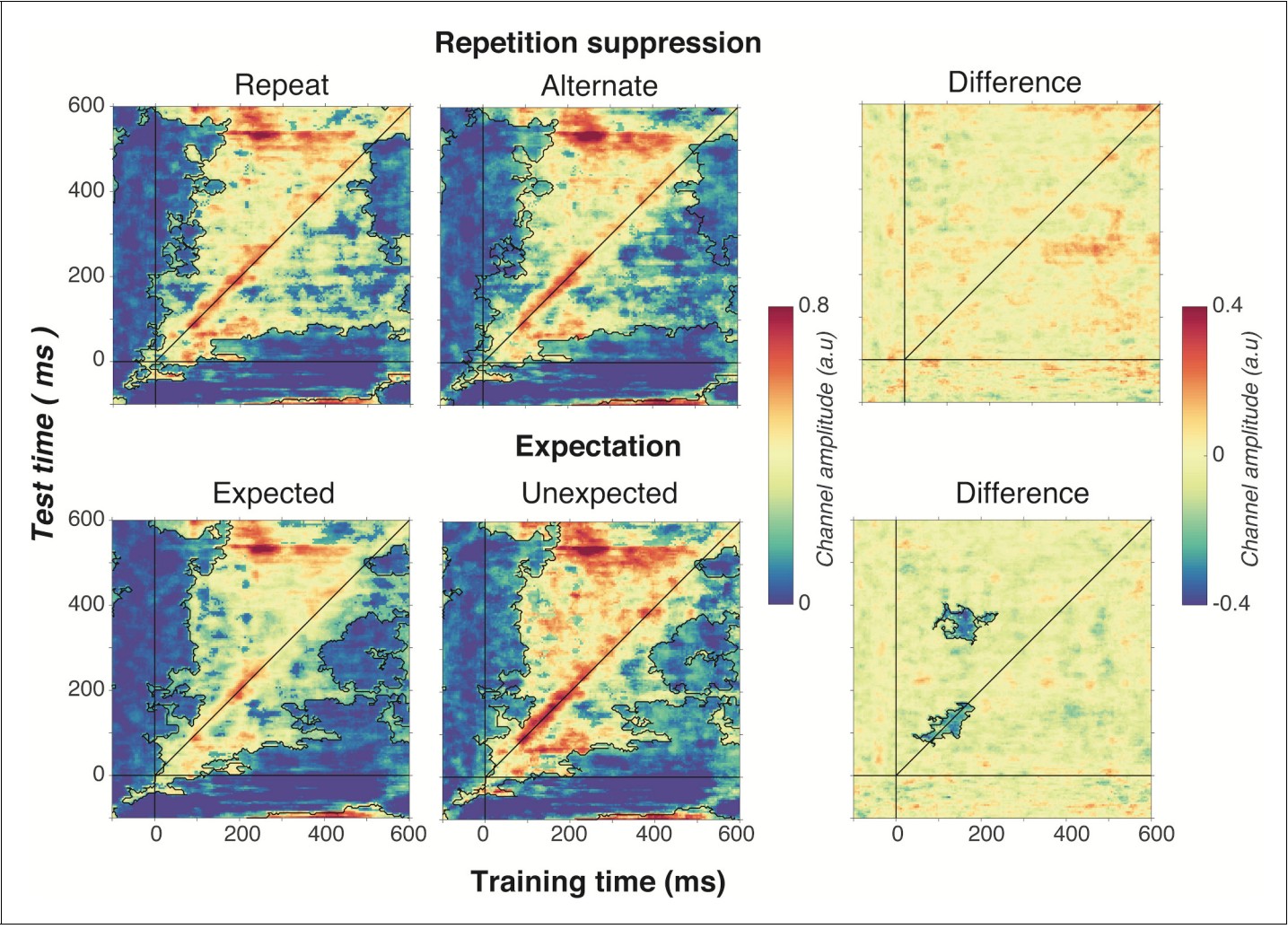

**Figure 6.** Cross-temporal generalisation of the forward encoding model based on grating orientations for the main effects of repetition suppression (upper panels) and expectation (lower panels). The maps have been thresholded (indicated by opacity) to show clusters (black outlines) of significant orientation selectivity (permutation testing, cluster threshold p < 0.05, corrected cluster statistic p < 0.05, 5000 permutations). The difference between the conditions is shown in the right-hand column (permutation testing, cluster threshold p < 0.05, corrected cluster statistic p < 0.05). Opacity and outlines indicate significant differences.

DOI: https://doi.org/10.7554/eLife.33123.009

have more information about an unexpected stimulus so that a correct response can be made. Our findings thus provide a novel insight into how predictive coding might change neural representations of sensory information.

The lack of evidence for sharpening of neural tuning in the current results is in contrast to the findings of a previous study (*Kok et al., 2012*), in which a high-level prediction error led to 'sharper' multivariate decoding for expected versus unexpected visual stimuli. In their study, *Kok et al. (2012)* used an auditory tone to cue the orientation of a subsequent visual stimulus, and found significantly reduced off-label classification accuracy for predicted than for unpredicted stimuli. They concluded that predictions cause sharpening of stimulus representations. More recently, using the same task combined with a forward encoding approach, *Kok et al. (2017)* showed that response gain is increased for a predicted stimulus.

It is natural to ask why the results of the current study differ from those of Kok and colleagues outlined above. One possible explanation lies in the different approaches used to generate expectations across the studies. Specifically, whereas Kok et al. manipulated expectations by pairing an auditory cue with a visual stimulus, we exploited the properties of the visual stimuli themselves (i.e. their

orientation) to generate expectations within blocks of trials. An intriguing possibility is that predictions requiring integration of sensory events from two or more modalities lead to increased sharpening, whereas predictions made within a single sensory modality lead to decreased gain. This might in turn relate to the noted differences between simple 'local' and higher-order 'global' type predictions (*Bekinschtein et al., 2009*; *King et al., 2014*), which lead to distinct patterns of stimulus-selective decoding. A similar discrepancy relating to the effects of attention on sensory representations has been widely discussed, with some studies finding sharpening of stimulus representations with attention, and others showing gain enhancement (*Liu et al., 2007*; *Maunsell, 2015*; *Maunsell and Treue, 2006*; *Treue and Martínez Trujillo, 1999*). The differences between these results may potentially have arisen because the tasks relied upon different types of attention (e.g., spatial versus feature-based). Future studies could determine whether this same divergence occurs for prediction effects.

The current work applied multivariate model-based approaches to EEG data to determine how prediction and repetition suppression affect neural representations of perceptual information. We chose to use EEG so we could recover the temporal dynamics of these effects – something that would not be possible with the BOLD signal used in fMRI – and because EEG is the most widely used tool for measuring expectation effects in human participants (see *Garrido et al., 2009* and *Paavilainen, 2013* for review), thus facilitating comparison of our findings with those of other studies. We estimated orientation selectivity using all EEG electrodes distributed across the scalp for two principal reasons. First, we wanted to limit experimenter degrees of freedom (*Simmons et al., 2011*) potentially introduced through the post-hoc selection of subsets of electrodes. Second, given the broad spatial resolution of EEG, we reasoned that activity recorded from electrodes at any given scalp location could potentially carry important feature-selective information from a number of neural sources. The results revealed that orientation-selective information appears largely driven by electrodes over occipital-parietal regions (*Figure 3D*), consistent with a number of previous studies that employed visual decoding of M/EEG data (*Cichy et al., 2014*; *Cichy et al., 2015*; *Stokes et al., 2015*). As noted above, however, it is entirely possible that the effects we observed here arose from sources well beyond the occipital and parietal regions, or even potentially outside the visual cortical hierarchy. Limitations in the temporal and spatial resolution of current human imaging methods make it impossible to pinpoint the timing and location of interactions between visual areas that might reflect the cascade of predictions and prediction errors involved in sensory encoding. By combining the current paradigm and multivariate modelling with invasive recordings in animal models – for example, using calcium imaging or extracellular electrode recordings – it should be possible to test some of the key claims of predictive coding theory that we have examined here, but at the level of individual neurons.

Surprisingly, few studies have used invasive recording methods to test how predictive coding affects stimulus representations at the neuronal level. One study in macaques (*Kaliukhovich and Vogels, 2011*) used a design similar to that of Summerfield and colleagues, but with high-level objects (fractals and real-world objects) as stimuli. That study found that expectation did not attenuate repetition suppression in either spiking activity or local field potentials within the inferior temporal cortex. A later fMRI study in humans (*Kovács et al., 2013*) used a similar stimulus set, and also found no attenuation of repetition suppression by expectation in the same cortical region. A follow-up study provided a potential explanation for these findings by showing that the attenuation of neural responses associated with repetition suppression is found with familiar stimuli, but not with unfamiliar stimuli (*Grotheer and Kovács, 2014*). Viewed in this light, the stimulus sets used by (*Kaliukhovich and Vogels, 2011*) might not have been sufficiently familiar to yield effects of expectation in their non-human primate observers.

Other work has shown that context plays an important role in determining the magnitude of neuronal responses to sensory events. Thus, for example *Ulanovsky et al. (2003)* found that rare auditory stimuli generate significantly larger responses in primary auditory cortical neurons than more commonly occurring stimuli. This result has been interpreted as a single-neuron analogue of the mismatch negativity, but the design used in the study did not control for adaptation effects, thus making it difficult to draw an unambiguous comparison with the current work. In the visual domain, oddball stimuli have also been found to modulate neuronal activity in rats, characterised by an enhancement of responses in the higher-order latero-intermediate area (*Vinken et al., 2017*). Moreover, *Fiser et al. (2016)* found that neurons in mouse primary visual cortex show a greater response

when task-irrelevant visual stimuli that had been presented during training were omitted, suggesting that an established expectation had been violated. This result is consistent with the literature on the mismatch negativity, in which the omission of an expected stimulus results in a large prediction error (*Garrido et al., 2009*; *Wacongne et al., 2011*). In non-human primates, neurons in the inferior temporal cortex show an enhanced response to unexpected relative to expected stimuli (*Kaposvari et al., 2018*), and population decoding accuracy is higher for unexpected compared with expected stimuli (*Kumar et al., 2017*). Critically, however, no study has simultaneously recorded neuronal activity in multiple cortical regions to determine whether predictions generated in one area refine responses in a second area, as postulated by predictive coding theory (*Friston, 2005*; *Rao and Ballard, 1999*). Such a direct demonstration is necessary to provide a strong test of the central notion that cortical areas pass signals between themselves in order to generate expectations.

Unlike the effects of expectation, there is a large body of electrophysiological work showing that sensory adaptation influences neuronal activity (*Adibi et al., 2013*; *Adibi et al., 2013*; *Felsen et al., 2002*; *Kohn and Movshon, 2004*; *Patterson et al., 2013*). For instance, there is a sharpening of stimulus selectivity in MT neurons following 40 s of adaptation to a drifting grating (*Kohn and Movshon, 2004*). As we have highlighted, however, prolonged adaptation is likely also associated with a significant prediction that the next stimulus will be the same as the previous one. Perhaps more relevant to the current results, *Patterson et al. (2013)* found that the width of orientation tuning in V1 is only marginally sharpened following brief (400 ms) periods of adaptation. Again, however, their study did not control for expectation, so it is impossible to determine the role of predictive coding in their observations. Our finding that repetition suppression did not affect the bandwidth of orientation selectivity measured using EEG is also consistent with models of orientation adaptation based on human psychophysical data, which suggest that adaptation does not affect the tuning width of the adapted neural populations (*Clifford, 2002*; *Clifford, 2014*; *Dickinson et al., 2010*; *Dickinson et al., 2017*; *Tang et al., 2015*).

In summary, we have shown that repetition suppression and expectation differentially affect the neural representation of simple, but fundamental, sensory features. Our results further highlight how the context in which a stimulus occurs, not just its features, affect the way it is represented by the brain. Our findings suggest encoding priority through increased gain might be given to unexpected events, which in turn could potentially speed behavioural responses. This prioritised representation is then re-activated at a later time, supporting the idea that feedback from higher cortical areas reactivates an initial sensory representation in early cortical areas.

## Materials and methods

### Participants

A group of 15 healthy adult volunteers (nine females, median age = 20.5 years, range = 18 to 37 years) participated in exchange for partial course credit or financial reimbursement (AUD$20/hr). We based our sample size on work that investigated the interaction between repetition suppression and prediction error (*Summerfield et al., 2008*), and that used forward encoding modelling to investigate orientation selectivity using MEG with a comparable number of trials as the current study (*Myers et al., 2015*). Each person provided written informed consent prior to participation, and had normal or corrected-to-normal vision. The study was approved by The University of Queensland Human Research Ethics Committee and was in accordance with the Declaration of Helsinki.

### Experimental setup

The experiment was conducted inside a dimly-illuminated room with the participants seated in a comfortable chair. The stimuli were displayed on a 22-inch LED monitor (resolution 1920 × 1080 pixels, refresh rate 120 Hz) using the PsychToolbox presentation software (*Brainard, 1997*; *Pelli, 1997*) for MATLAB (v7.3). Viewing distance was maintained at 45 cm using a chinrest, meaning the screen subtended 61.18° x 36.87° (each pixel 2.4′ x 2.4′).

## Visual task

The stimuli were Gabors (diameter: 5°, spatial frequency: 2 c/°, 100% contrast) presented centrally in pairs for 100 ms, separated by 500 ms (600 ms stimulus onset asynchrony) with a variable (650 to 750 ms) inter-stimulus interval between trials. Across the trials, the orientations of the Gabors were evenly spaced between 0° and 160° (in 20° steps) so we could reconstruct orientation selectivity contained within the EEG response using forward encoding modelling. The relationship of the orientations of the pairs Gabors was also used to construct the different repetition suppression and prediction conditions. The orientation presented in the second Gabor in the pair could either repeat or alternate with respect to the orientation of the first Gabor. In the alternation trials, the orientation of the first Gabor was drawn randomly, without replacement, from an even distribution of orientations that was different to the orientation of the second Gabor. To vary the degree of prediction, in half of the blocks 80% of the trials had repeated orientations and 20% of the trials had alternating orientations, whereas in the other half of the blocks these contingencies were reversed. This design allowed us to separately examine the effects of repetition suppression and prediction because of the orthogonal nature of the blocked design. The blocks of 135 trials (~3 mins) switched between the expectation of a repeating or alternating pattern, with the starting condition counterbalanced across participants.

The participants' task was to monitor the visual streams for rare, faintly coloured (red or green) Gabors and to discriminate the colour as quickly and accurately as possible. Any trial with a coloured target was excluded from analysis. The orientation match between the pairs was made to be consistent with the dominant contingency (i.e. repeated or alternating) within that block. Pilot testing was used prior to the main experiment to set the task at approximately threshold, to ensure that participants focused exclusively on the colour-discrimination task rather than the orientation contingencies associated with prediction and repetition. Only one participant reported being aware of the changing stimulus contingencies across the blocks when asked at the end of the experiment, and excluding this participant's data had no effect on the key results reported here. Self-paced breaks were provided between each of the 20 blocks within a session, at which time feedback was provided on performance in the preceding block. Each participant completed two sessions of 2700 trials each (5400 trials in total), with each session lasting around 70 min of experimental time and 45 min of EEG setup.

## EEG acquisition and pre-processing

Continuous EEG data were recorded using a BioSemi Active Two system (BioSemi, Amsterdam, Netherlands). The signal was digitised at 1024 Hz sampling rate with a 24-bit A/D conversion. The 64 active scalp Ag/AgCl electrodes were arranged according to the international standard 10–20 system for electrode placement (*Oostenveld and Praamstra, 2001*) using a nylon head cap. As per BioSemi system design, the common mode sense and driven right leg electrodes served as the ground, and all scalp electrodes were referenced to the common mode sense during recording.

Offline EEG pre-processing was performed using EEGLAB in accordance with best practice procedures (*Bigdely-Shamlo et al., 2015*; *Keil et al., 2014*). The data were initially down-sampled to 256 Hz and subjected to a 0.5 Hz high-pass filter to remove slow baseline drifts. Electrical line noise was removed using *clean_line.m,* and *clean_rawdata.m* in EEGLAB (*Delorme and Makeig, 2004*) was used to remove bad channels (identified using Artifact Subspace Reconstruction), which were then interpolated from the neighbouring electrodes. Data were then re-referenced to the common average before being epoched into segments around each stimulus pair (−0.5 s to 1.25 s from the first stimulus in the pair). Systematic artefacts from eye blinks, movements and muscle activity were identified using semi-automated procedures in the SASICA toolbox (*Chaumon et al., 2015*) and regressed out of the signal. After this stage, any trial with a peak voltage exceeding ±100 $uV$ was excluded from the analysis. The data were then baseline corrected to the mean EEG activity from −100 to 0 ms before the presentation of the second Gabor in the pair. Critically, the orientations of the first and second gratings were precisely balanced across the conditions to avoid any systematic bias in orientation information being carried forward by the first grating within each pair. Specifically, for every unexpected stimulus presented in the second grating there was an equal number of every other orientation that was expected to be presented. As the analysis we employed used a

regression-based approach, any carry over of orientation-selective information from the first to the second grating therefore could not systematically bias the results.

## Experimental design

We used a modified version of a factorial design that has previously been used to separately examine the effects of repetition suppression and prediction error (*Kaliukhovich and Vogels, 2011*; *Kovács et al., 2013*; *Summerfield et al., 2008*; *Summerfield et al., 2011*; *Todorovic et al., 2011*; *Todorovic and de Lange, 2012*). By comparing the two repeat conditions with the two alternating conditions, we could examine repetition suppression while controlling for different levels of expectation. Conversely, by comparing across the expected and unexpected trials, we could examine prediction error while controlling for repetition suppressi.

The relationship between the pairs of orientations for the different expectation conditions was based on the original study (*Summerfield et al., 2008*), and on other studies (*Kaliukhovich and Vogels, 2011*; *Kovács et al., 2013*) that examined the interaction between repetition suppression and expectation. In the repeating condition, the orientation of the second Gabor is expected to be the same as the orientation of the first, whereas in the alternating condition the orientation of the second Gabor is expected to be *different* from that of the first. This relationship between the expected orientations of the stimuli in the alternating condition is slightly different to another modification of the paradigm which employed a more limited range of stimuli (*Todorovic et al., 2011*; *Todorovic and de Lange, 2012*). Specifically, the paradigm introduced by Todorovic and colleagues used two or three auditory tones of different frequencies. In their alternating condition, the expectation was that one tone would follow another (i.e. 1000 Hz and then 1032 Hz); this was then violated when a 1000 Hz tone was repeated. In this paradigm, an exact frequency was expected in the alternating condition, a design feature that differs from the paradigm used in the current work where there was no specific expectation of the orientation of the second Gabor based on the orientation of the first in the alternating condition. Instead the expectation in the alternating condition was that the orientation would change, and this could be violated by repeating the orientation. In this sense, there was no specific expectation about the second orientation in the alternating condition. Instead, the rule concerened the alternation or repetition of the first orientation. We did not implement the Todorovic et al. paradigm because of the combinatorial explosion of stimulus conditions it would

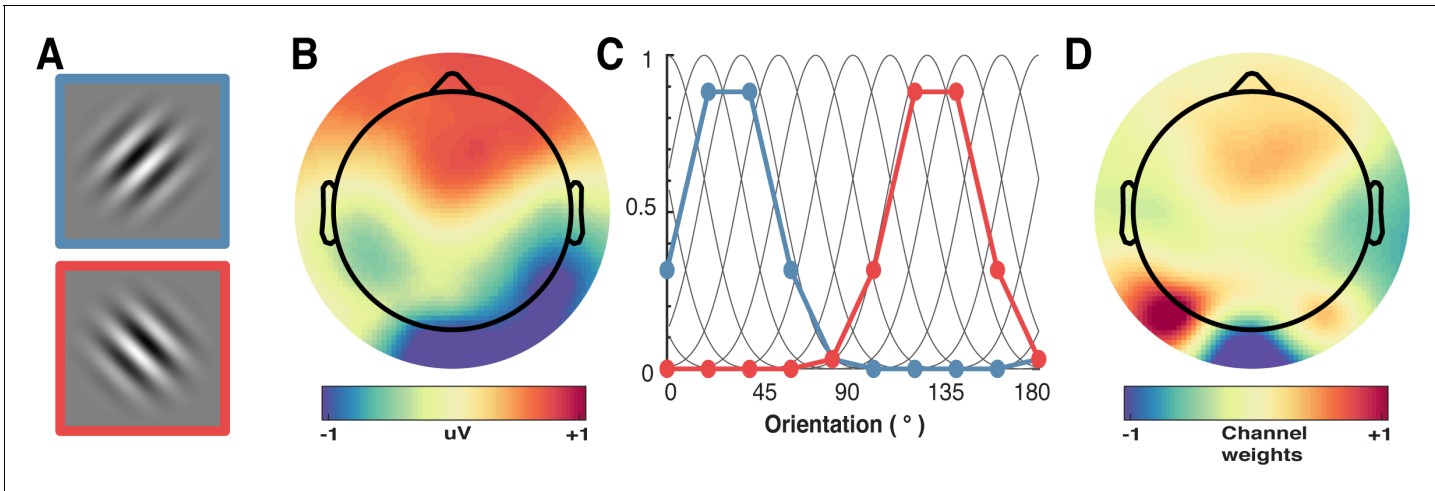

**Figure 7.** A schematic of the forward-encoding approach applied to EEG activity. (**A**) Participants viewed individual gratings at fixation, each with a specific orientation. (**B**) Neural activity evoked by each grating was measured over the entire scalp. (**C**) Evoked neural responses were convolved with canonical orientation-selective functions (grey lines in C) to determine coefficients for the different orientations (coloured dots and lines, which match the colours of the outlined gratings in A). These coefficients were then used to generate a regression matrix. (**D**) General linear modelling was used on a subset of training trials to generate weights for each channel. These weights were inverted and simultaneously applied to an independent test set of data to recover orientation selectivity in the EEG activity. As EEG activity has high temporal resolution, we can apply the procedure to many epochs following stimulus presentation to determine the temporal dynamics of orientation processing (see *Figure 3*).
DOI: https://doi.org/10.7554/eLife.33123.010

require to measure orientation selectivity (such that every orientation is predicted by another orientation). Future work could investigate how this subtle change in paradigm design affects the encoding of stimulus information.

## Forward encoding modelling

We used a forward encoding approach to estimate the amount of orientation-selective information contained in the EEG data at each time point of the trial. This approach differs from standard decoding approaches by modelling each presented orientation as a continuous variable of a set of tuned orientation-selective channels. The forward-encoding technique has been successfully used to reconstruct colour (*Brouwer and Heeger, 2009*), spatial (*Sprague and Serences, 2013*) and orientation (*Ester et al., 2016*) selectivity in fMRI data. More recently, the same approach has been applied to EEG and MEG data, which have inherently better temporal resolution than fMRI (*Garcia et al., 2013*; *Kok et al., 2017*; *Myers et al., 2015*; *Wolff et al., 2017*).

We applied forward encoding modelling to determine how repetition suppression and prediction error affected orientation selectivity. To do this, the second orientation (*Figure 7A*) in the Gabor pair in each trial was used to construct a regression matrix, with separate regressors for the nine orientations used across the experiment. This regression matrix was convolved with a set of basis functions (half cosines raised to the $8^{th}$ power (*Figure 7C*), which allowed complete and unbiased coverage of orientation space) to allow us to pool similar information patterns across nearby orientations (*Brouwer and Heeger, 2009*). We used this tuned regression matrix to estimate time-resolved orientation selectivity contained within the EEG activity in a 16 ms sliding window, in 4 ms steps (*Figure 7B*; *Myers et al., 2015*). To avoid overfitting, we used a leave-one-out cross-validation procedure where the regression weights were estimated for a training set and applied to an independent test set (*Figure 7D*). All trial types (including target trials) were used in training and test sets. This was done by solving the linear equation:

$$B_1 = WC_1 \tag{1}$$

where $B_1$ (64 sensors x N training trials) is the electrode data for the training set, $C_1$ (nine channels x N training trials) is the tuned channel response across the training trials, and W is the weight matrix for the sensors we want to estimate (64 sensors x nine channels). W can be estimated using least square regression to solve *equation (2)*:

$$W = (C_1 C_1^T)^{-1} C_1^T B_1 \tag{2}$$

The channel response in the test set $C_2$ (nine channels x N test trials) was estimated using the weights in (2) and applied to activity in $B_2$ (64 sensors x N test trials).

$$C^2 = (WW^T) W^T B^2 \tag{3}$$

We repeated this process by holding one trial out as test, and training on the remaining trials until all trials had been used in test and training. The procedure was repeated for each trial within the trial epoch. We then shifted all trials to a common orientation, meaning that 0° corresponded to the orientation presented on each trial.

The reconstructed channel activations were separated into the four conditions, and averaged over trials. These responses were then smoothed with a Gaussian kernel with a 16 ms window, and fitted with a Gaussian function (4) using non-linear least square regression to quantify the amount of orientation selective activity.

$$G(x) = A \, exp\left(-\frac{(x-\phi)^2}{2\sigma^2}\right) + C \tag{4}$$

Where *A* is the amplitude representing the amount of orientation selective activity, $\phi$ is the orientation the function is centred on (in degrees), $\sigma$ is the width (degrees) and *C* is a constant used to account for non-orientation selective baseline shifts.

## Multivariate pattern analysis

We conducted a multivariate pattern analysis to build upon the initial forward encoding results which showed that unexpected stimuli elicit greater orientation selectivity than expected stimuli. This analysis used the same data as the forward encoding analysis. We used the *classify* function from *Matlab 2017a* with the 'diaglinear' option to implement a Naive Bayes classifier. For each time point, we used the same cross-validation procedure as the forward encoding modelling with the same averaging procedure to select train and test sets of data. The classifier was given the orientations of the training data and predicted the orientation of the test data. A trial was labelled correct if the presented orientation was produced. To facilitate comparison of the results with those of *Kok et al. (2012)*, we found the peak classification accuracy for each participant in the 600 ms following stimulus presentation. The same wide time window was used across conditions to accommodate large inter-individual differences in peak classification without biasing the results toward one particular condition.

## Statistical testing

A non-parametric sign permutation test was used to determine the null distribution for testing (*Wolff et al., 2017*). This method makes no assumptions about the underlying shape of the null distribution. This was done by randomly flipping the sign of the data for the participants with equal probability. Fifty thousand (50,000) permutations were used for the time-series data, whereas only 5000 were used for the temporal generalisation plots because of the significantly greater computational demands.

Cluster-based non-parametric correction (50,000 permutations for timeseries and 5000 for temporal generalisation) was used to account for multiple comparisons, and determined whether there were statistical differences between the contrasting conditions. Paired-samples t-tests were used to follow up the analysis in *Figure 4* within a specified time window, and no correction was applied. A two-way repeated measures ANOVA (implemented using GraphPad Prism 7.0 c, La Jolla, CA) was used to analyse the multivariate pattern analysis results shown in *Figure 5*.

## Acknowledgements

This work was supported by the Australian Research Council (ARC) Centre of Excellence for Integrative Brain Function (ARC Centre Grant CE140100007) to JBM and EA, and by an ARC Discovery Project (DP170100908) to EA. JBM was supported by ARC Australian Laureate Fellowship (FL110100103).

## Additional information

### Funding

| Funder | Grant reference number | Author |
| --- | --- | --- |
| Australian Research Council | CE140100007 | Ehsan Arabzadeh<br>Jason B Mattingley |
| Australian Research Council | DP170100908 | Ehsan Arabzadeh |
| Australian Research Council | FL110100103 | Jason B Mattingley |

The funders had no role in study design, data collection and interpretation, or the decision to submit the work for publication.

### Author contributions

Matthew F Tang, Conceptualization, Formal analysis, Writing—original draft, Writing—review and editing; Cooper A Smout, Ehsan Arabzadeh, Jason B Mattingley, Conceptualization, Writing—review and editing

## Author ORCIDs

Matthew F Tang http://orcid.org/0000-0001-5858-5126
Cooper A Smout http://orcid.org/0000-0003-1144-3272
Ehsan Arabzadeh http://orcid.org/0000-0001-9632-0735
Jason B Mattingley http://orcid.org/0000-0003-0929-9216

## Ethics

Human subjects: Each participant provided written informed consent prior to participation. The study was approved by The University of Queensland Human Research Ethics Committee (approval number 2012000392) and was in accordance with the Declaration of Helsinki

## Decision letter and Author response

Decision letter https://doi.org/10.7554/eLife.33123.016
Author response https://doi.org/10.7554/eLife.33123.017

## Additional files

### Supplementary files

• Transparent reporting form
DOI: https://doi.org/10.7554/eLife.33123.011

### Data availability

The EEG data have been deposited on Dryad 10.5061/dryad.3d7kq

The following dataset was generated:

| Author(s) | Year | Dataset title | Dataset URL | Database and Identifier |
|---|---|---|---|---|
| Tang M, Smout C, Arabzadeh E, Mattingley J | 2018 | Data from: Prediction Error and Repetition Suppression Have Distinct Effects on Neural Representations of Visual Information | https://dx.doi.org/10.5061/dryad.3d7kq | Dryad Digital Repository, 10.5061/dryad.3d7kq |

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
