## [Decision Letter]

Thank you for submitting your article "Prediction Error and Repetition Suppression Have Distinct Effects on Neural Representations of Visual Information" for consideration by *eLife*. Your article has been reviewed by Timothy Behrens as the Senior Editor, a Reviewing Editor, and three reviewers. The following individuals involved in review of your submission have agreed to reveal their identity: Peter Kok (Reviewer #1); Hamed Nili (Reviewer #3).

The reviewers have discussed the reviews with one another and the Reviewing Editor has drafted this decision to help you prepare a revised submission.

All the reviewers expressed enthusiasm for your work and recognised the quality of the study and the potential import of the findings. Their concerns focused on a number of issues that would need to be addressed for a revised manuscript to be acceptable to *eLife*.

1) The details of several of the analyses were unclear, as noted by all three reviewers. In particular, exactly what was used for training and what for test was unclear for some of the later figures. There is also a seeming discrepancy between Figure 2C and the text. Like reviewer 1, I was surprised by the form of the ERPs in Figure 2; please clarify how these were obtained, if not by standard methods. Reviewer 2 also questions the interpretation of the results presented in Figure 7; it would be important to address that at revision.

2) The reviewers noted that the Introduction and Discussion section were at times misaligned with the results themselves. In particular, the allusion to predictive coding seemed to be quite loose in places; please make clear which results support predictive coding, and which do not. It would be important for this work to be discussed in the context of previous studies using ERPs/MEG in conjunction with this paradigm, such as Kok et al., 2017. The results described here seem very different from those in that paper (and also discrepant with de Gardelle et al., 2011); the reviewers ask that you at minimum acknowledge and discuss these inconsistencies.

3) Reviewer 2 raises a key issue about the provenance of the neural signals. Are you able to offer any reassurance that these signals are really "perceptual" in origin?

4) Reviewer 3 asks how the decoding varies with the interaction of the two factors. This seems like a potentially important point that is unaddressed in the current version.

5) Please also take care to address the other minor points raised by the reviewers.

Reviewer #1:

This study by Tang et al., investigates the effects of expectation and repetition on the neural processing of visual stimuli. The authors use both conventional EEG amplitude measurements, as well as forward model decoding to study the neural representations of the stimuli. They find that unexpected stimuli are represented substantially better in the neural signal than expected ones – an intriguing result. This is an interesting topic, investigated using excellent methods that seem executed well. I do have some questions, many of which are for clarification.

1) The authors discuss a prediction error (Expected < Unexpected) effect, but from Figure 2C it seems that expected stimuli (blue) evoke more activity than unexpected ones (red). Please explain.

2) I am not an EEG expert, so please excuse me for my naivety, but I would have expected to see ERPs to consist of early components of alternating polarity, i.e. P1-N1-P2-N2-P3. However, here the amplitude of the EEG signal is only ever positive post-stimulus – is this the result of a specific aspect of the paradigm, or the preprocessing of the data?

3) When studying the effect of repetition on orientation-selective signals, can the authors distinguish whether any modulations would be the result of actual adaptation/repetition suppression, versus lingering activity evoked by the first grating affecting the decoding? Of course, neural adaptation in early sensory regions is the thing the authors are interested in – but imagine that the first grating is still being processed in, for instance, parietal and frontal areas at the time the second grating is presented. Then the decoder trying to decode the second grating would pick up, and be influenced by, these parietal and frontal signals evoked by the first grating, even though they do not reflect 'true' repetition suppression or adaptation. Is this a concern?

4) In Figure 6, there is a significant off-axis difference in decoding between expected and unexpected trials, which is very interesting. However, I'm having trouble interpreting the fact that this cluster appears above (rather than below, or both below and above) the diagonal. Which axis reflects 'training time' and which 'testing time'?

5) Figure 7 requires some clarification. In the text, the authors say "Here, we trained the forward encoding model on the orientation of the Gabor that was actually presented (stimulus driven), or on the orientation that was expected based on the first Gabor in the pair (non-stimulus driven)." Which one was true for Figure 7; trained on the presented or expected orientation? Or did this different between quadrants of the matrices? From the text, it seems to me that the decoder is always trained on the T1 orientations – however, if this is true, I don't understand why there is such strong decoder for both unexpected and expected gratings in the off-diagonal quadrants, given that we're only looking at the alternation trials here. That is, given that the T2 orientation different from the T1 orientation, how come it's possible to train on T1 (i.e. negative training times) and decode T2 (positive testing times)? Of course, an argument could be made for the unexpected alternation condition, given that there T1 is expected to repeat, but this is not true for the expected alternation condition.

6) "More interestingly, in the unexpected alternation condition there was significantly better on-axis orientation selectivity for the T1 orientation between 150 and 300 ms after the onset of T2, relative to the expected alternation condition (upper right quadrant of right panel in Figure 7)." So, if I understand correctly, in the post-T2 time window, there is both increased evidence for the T1 orientation (Figure 7), as well as for the T2 orientation (Figure 6), for unexpected compared to expected gratings? This is quite interesting, and surprising. Can the authors confirm that this interpretation of their effects is correct?

7) Was the decoder always trained on an equal mixture of expected and unexpected and repeating and alternating trials, to make sure the decoder was not biased towards one of these categories? For instance, given that expected gratings are much more numerous than unexpected ones, the decoder may be better at decoding expected trials if the trial counts are not balanced during training.

8) "It is noteworthy, however, that in their study Kok et al., (2012) employed a 'backward' decoding analysis to quantify sharpness, rather than forward encoding as here, which might account for the discrepant findings." I'm not sure I follow this reasoning, given that the authors replicate their own findings using a 'backward' decoding analysis as well. Also note that an MEG study using the same paradigm (Kok et al., 2017) did use forward modelling, and also found improved decoding for expected vs. unexpected gratings.

9) "An intriguing possibility is that combining predictions generated across distinct cortical areas (e.g., visual and auditory) leads to sharpening of tuning, whereas predictions generated within a single cortical area lead to gain modulation." This in indeed an intriguing possibility. I would like to note, however, that it seems unlikely that the predictions in the current study are generated in a single cortical region (e.g. V1), given their complexity. After all, it is not simply grating 1 that predicts the orientation of grating 2, but the orientation of grating 1 in conjunction with the current context ("am I in a repetition or alternation block?"). This type of context seems unlikely to be encoded in early visual cortex but may perhaps be signalled by higher level regions in frontal cortex (or elsewhere) instead.

10) "The data were then baseline corrected to the average EEG activity from -100 to 0 ms before the presentation of the second Gabor pair." The phrasing is unclear. Does this mean that activity was baseline before presentation of the first or the second member of each pair?

Reviewer #2:

Tang et al., present an EEG study in which they investigate how prediction error and repetition suppression affect neural representations of visual information. They manipulated two factors: (i) whether two stimuli repeated in a trial or not; (ii) whether a repeat was very likely to occur or not. The authors performed a forward encoding analysis to characterize orientation selectivity. They show that repetition suppression decreases overall responsiveness but does not affect orientation selectivity. In contrast, unexpected stimuli, when compared to expected stimuli, only showed a minor difference in overall responsiveness, but instead seemed to be associated with a larger gain in orientation tuning curves without a change in tuning width.

This study addresses important questions and provides relevant findings on these issues. The experimental design and implementation is done sufficiently well, the analyses are well thought off, and the manuscript is overall clearly written.

Nevertheless, several major revisions will be necessary before the actual implications of the findings are clear. I will first go into a few important concerns.

First, it is unclear to me which neural signal underlies the orientation selectivity that is measured. Even if we would accept that it is a signal from visual regions, it must be very far removed from the responses of single neurons or orientation columns. What are the different possibilities, and what are the implications of this uncertainty for our understanding of the neural correlates of prediction error processing? However, before such questions are addressed, it first has to be clarified to what extent the findings relate to visual processing. If I understand correctly, the forward encoding modelling to estimate orientation selectivity considers all electrodes across the entire brain. Therefore, it is impossible to determine which electrodes contribute to the orientation selectivity and its modulation by expectations. Is it possible to perform further analyses to clarify this? Imagine that the expectation effect would be largely driven by frontal electrodes, then the results would not necessarily tell us much about perceptual processing.

Second, and partially related to the first, subsection “Repetition suppression and prediction error affect the overall level of neural activity” and Figure 2 show that repetition and expectation effects are found both in occipital-parietal and in frontal electrodes. The effect for repetitions in occipital-parietal electrodes (repeat < alternation) is in the expected direction. However, the expectation effect seems to be in the wrong direction in occipital-parietal electrodes (expected > unexpected). From a predictive coding perspective, unexpected repetitions/alternations should result in a higher prediction error and thus higher response. Just like alternation trials result in a higher response. How can we interpret the further multivariate results if this main effect of expectation is hard to reconcile with what we see for repetition suppression and with what we would expect based on e.g. the fMRI experiment of Summerfield et al.,?

Third, and again somewhat related to the previous concern, at many points the relationship between prediction errors, repetition suppression, and neuronal adaptation is not characterized well. A few examples:

In the Abstract prediction errors are claimed to provide a possible explanation of repetition suppression. However, in previous work (as well as in this study), expectation effects are dissociated from repetition suppression. The latter can be explained perfectly well by stimulus driven adaptation effects. These expectation effects are smaller and less robust than stimulus driven adaptation, suggesting a minor modulatory role of perceptual expectation.

Introduction: "Here we asked whether a predictive coding theory can account for the changes in neural representations observed with repetition suppression." How is this question answered?

Fourth, the authors should adjust their claims and interpretation to the strength of the evidence provided, from abstract to Discussion section. First, in order to do this, the statistical reporting should be more specific. The authors should report actual p-values (of course rounded to a reasonable precision), instead of arbitrary thresholds, such as p<.025 or p<.05. Also, for the statistics shown mostly in figures, such as Figure 6 and Figure 7, it would be important to mention the highest p-value obtained, and not just say p<0.05. Most effects seem to be close to 0.05, which should be reflected upon. One p-value signals a very low probability (p =.0049; subsection “Prediction errors increase the amount of orientation-selective information contained within patterns of EEG activity”), however, it is the result of a circular analysis. The time window on which this analyses was done was pre-selected based on a statistical test on the same data. This should be indicated. Overall, the amount of evidence presented in this paper seems sufficient for a publication but not strong enough to just pass without further discussion and adjusting the interpretations to the strength of the evidence.

Fifth, the orientation selectivity is determined and tested using the second gabor in the trials. If you find a difference between two conditions A and B, with A showing better tuning than B, how can you know whether it is A that has an increased tuning or B a decreased tuning? To me it seems that the orientation selectivity on the first gabor in the trials would be a good baseline/reference. So why did you not include the first gabor in the trials in your orientation selectivity analyses? You conclude "prediction error was associated with a significantly increased orientation-selective response through a gain modulation soon after the stimulus was presented", but how can you differentiate this from "The absence of prediction error was associated with a significantly decreased orientation-selective response through a gain modulation soon after the stimulus was presented". When split in subsets of trials according to the relationship with the second stimulus, these analyses on the first gabor would also serve as a useful statistical control for the analyses presented in Figure 2. If the statistical thresholding is stringent enough, then doing the same analysis on the first stimulus should not give false positives.

Reviewer #3:

I found the paper very interesting.

While they show that the univariate effects of repetition and expectation are comparable (same areas, different sign), they show that while repetition doesn't affect the selectivity of representations, expectation leads to an increase in selectivity by gain modulation.

Given the importance of the multivariate analysis in directly leading to the main claim of the paper, I have a few comments on it:

1) The authors use an encoding approach to test for information processing. That is very clearly explained in the Materials and methods section. At the last step, they fit a Gaussian (Equation 4), which has 4 parameters. So, for example Figure 3C quantifies the orientation selectivity by parameter A. Now whilst that is an interesting fact, it might be the case that there are other differences in the two (e.g. that σ is smaller in the repeated than the non-repeated).

2) The multivariate analysis that compares orientation selectivity in repeat/ alternate and expected/unexpected is key to the claims of the paper. However, the authors could have looked at all 4 possible combinations (similar to what they do for the LDA visualized in Figure 5). For example, it would be interesting to test if orientation selectivity in the expected is different when the stimulus is repeated or alternated. Figure 4 would be a good summary if there was no interaction between the two at any time point.

3) Figure 5 is an interesting control analysis. However, there is no information on how they performed cross validation, what time points are averaged to give the result and how stable the results are!

4) Although I find the cross-temporal analysis on the weights from the forward encoding model interesting, I think that some speculations could be directly tested with pattern similarity analysis. For example, for the unexpected stimulus, they could directly test whether the representation of the unexpected is reactivated at 300-400 ms. They could do non-parametric statistics for the inference. I also make the same suggestion for analysis of Figure 7.

[Editors’ note: this article was subsequently rejected after discussions between the reviewers, but the authors were invited to resubmit after an appeal against the decision.]

Thank you for submitting your revised submission entitled "Prediction Error and Repetition Suppression Have Distinct Effects on Neural Representations of Visual Information" for consideration by *eLife*. Your revised article has been reviewed by three peer reviewers, and the evaluation has been overseen by a Reviewing Editor and a Senior Editor. The following individuals involved in review of your submission have agreed to reveal their identity: Peter Kok (Reviewer #1); Hamed Nili (Reviewer #3).

Our decision has been reached after extensive consultation between the reviewers. Based on these discussions and the individual reviews below, we regret to inform you that we have decided to reject your submission in its current form.

As you can see from the comments below, all of the reviewers – and in particular #1 and #2 – felt that their comments had been insufficiently addressed at revision. The reviewing editor has read your revised manuscript and rebuttal with care and agrees. He has flagged the following major issues as follows:

(i) One major premise of the manuscript is that you claim to see a univariate effect of unexpected > expected trials. However, the plot you present in Figure 2 shows precisely the converse. The reviewers found your argument that this effect reverses over more anterior sites (not shown) unconvincing, particularly given that the major focus of the paper is on predictive coding in hierarchical sensory systems. This, coupled with the unconventional presentation of the ERPs (which do not resemble potentials that the reader would typically expect to see over visual regions), led the reviewers to doubt the claims made with regard to this signal.

(ii) Reviewers #1 and #2 raised substantive issues about the train/test approach that you took for the multivariate analyses. The paradigm you have employed allows the viewer to form expectations about the second grating on the basis of the first. This means that the neural signal elicited by the first grating may partly encode an expectation of the second grating, and the neural signal elicited by the second grating might partly include carryover from the first grating. Uncertainty over whether the effects you described have a straightforward interpretation was exacerbated by the asymmetry in the cross-validation plots. The reviewers remain unconvinced by your rather cursory replies to their questions regarding this point.

(iii) The reviewers still feel that the link to predictive coding is very loose. The manuscript continues to be motivated by the predictive coding framework, even though several of your results seem to directly contradict what might be expected under this theory. The reviewers found the exegesis to be rather contradictory, and the theoretical claims confusing in the light of the results.

(iv) There are continued concerns from reviewers #2 and #3 that your analyses are at least partly circular, and that there may be oddities introduced by your baselining techniques (reviewer #1).

Together, this prompted the decision to reject the manuscript as it stands. If you would like to appeal this decision, you are welcome to do so, but I would suggest that it is unlikely that the reviewers will be sympathetic to an appeal without a serious revision to the paper that addresses their points in full. For my part, I think that this manuscript has promise, but I feel that there are simply too many inconsistencies in the current version to warrant publication in *eLife*.

*Reviewer #1:*

Unfortunately, I feel that not all concerns have been satisfactorily addressed. In its current form, I am not fully confident that the results described reflect effects of expectation on perceptual representations. The authors' responses to my concerns expressed in the previous round have not fully taken away these concerns, as I detail below.

Original point 1. I'm still not quite sure how to interpret the amplitude results displayed in Figure 2C-D. It seems to me, from Figure 2C, that expected stimuli evoke a larger response than unexpected ones, yet the authors interpret this as a prediction error effect, with a larger neural response to unexpected stimuli (over frontal electrodes). In their response, they state that there was "more negativity associated with the unexpected stimulus", but I'm not sure how this follows from Figure 2C-D.

Original point 2. According to the authors, the fact that the ERP waveforms is (almost) only positive, rather than a sequence of positive and negative components, is the result of the minimal preprocessing applied (only a 0.5 Hz high-pass filter). However, from my limited experience with EEG, I would expect positive and negative components even with minimal preprocessing. I'm still somewhat confused by this.

Original point 5. I still don't understand why there is such strong off-diagonal decoding in Figure 7, given that the orientation of the second grating was randomized, as the authors state in the legend. In the orientation of the second grating was randomized, how is it possible to decode it when training on the first grating?

In response to my original point #10, the authors explain that the EEG data were baseline corrected to the interval -100 to 0 ms before presentation of the second Gabor in the pair. Isn't this problematic, in a paradigm in which grating 1 can lead to a prediction about grating 2? If there is an expectation signal present just before the appearance of grating 2 (as is quite likely), this would 'contaminate' the baseline, and effects found elsewhere (before -100 ms or after 0 ms) might actually reflect a negative image of expectation effects present in the pre-stimulus 2 interval.

Reviewer #2:

I thank the authors for their revision, which addresses some of the earlier concerns.

There are a few remaining issues, addressing them would improve the manuscript further.

About the localization of the signal:

Central to predictive coding theories is the idea that sensory predictions which are generated higher up in the hierarchy modulate responses in lower levels of sensory cortices. This statement is much more specific than the more general idea that: recent experience establishes expectations in the brain, which manifest somewhere and somehow in the encoding of sensory input. In the current state of the manuscript, this important nuance seems to be missing. Given that predictive coding theories have very specific hierarchical predictions, the argument that the authors wanted to "limit experimenter degrees of freedom potentially introduced through the post-hoc selection of subsets of electrodes" is simply not valid here. This is not a trivial issue, because as long as we do not constrain a theory, it cannot be falsified.

I understand that the authors might not want to or might not be able to localize their effects of expectation, but that limits the extent to which these findings are instructive about predictive coding theory and more generally about how the brain processes unexpected sensory events. Even if the general orientation selectivity is mainly determined by the signal at occipital and parietal electrodes, there is no guarantee that any interaction between expectation and orientation selectivity originates from the same signal in these electrodes. For example, I wonder whether a potential explanation for the discrepancy with Kok et al.'s work could be that their effects were localized to visual cortex? Perhaps in this study their expectation effects do not originate from in visual cortex, but for example only in frontal areas?

Without such information, the findings of this study reduce to: (a) that repetition suppression and expectation effects are dissociable (which already has been demonstrated by for example Todorovic and de Lange, 2012) and (b) expectation increases selectivity for (Gabor) stimuli somewhere in the brain for this particular task. The fact that this would be hard to reconcile with other findings in early visual cortex is then explained away by the argument that "as this is a relatively complex phenomenon it is likely to yield different outcomes when different levels of prediction are manipulated". This reasoning does not echo the idea of predictive coding as universal principle of cortical responses. A good theory should be able to make general predictions that are valid in a (relatively) broad range of situations, without requiring ad hoc explanations for different experiments.

The authors should be more explicit about these problems.

Specific points:

"One mechanism by which the brain reduces its information processing load is to encode successive presentations of the same stimulus in a more efficient form, a process known as neural adaptation": The authors have added references for this sentence, but I think they still can't justify the statement. Right now, the sentence reads like the function of adaptation is a settled case. Given our current lack of understanding of the mechanisms underlying adaptation, any suggestions about its function remain highly speculative. I suggest you change this sentence to "Neural adaptation is one mechanism by which the brain might reduce its information…".

In their rebuttal the authors confirm that they find Expected > Unexpected in occipital-parietal electrodes (while Expected < Unexpected in central frontal electrodes). This is a finding that contradicts the hallmark of predictive coding theory, namely that neural response strength signals prediction error. The authors should address this discrepancy. On the other hand, the authors still use the term "prediction errors" in reference to their results. For example, in the Discussion section they say: "We found that prediction errors were associated with increased gain of stimulus representations…". Yet, as they acknowledge, they can't know where the change in stimulus presentations took place, while they only find prediction errors in a frontal cluster.

Comment 5:

I understand that care should be taken for these comparisons, but still it is very valid to compare with the first stimulus. Take the field of single-unit animal studies of repetition suppression, would they not always show data/analyses for the first stimulus also? I think it is an important statistical control for various effects (as I mentioned in my original email).

Reviewer #3:

I think the revised document is much more clear and transparent.

After reading the responses and details of Figure 5, I have a slight concern that it might suffer from some circularity.

The figure is very clear and the results are consistent. However, the time period for which these results are displayed can affect the trend.

Ideally one would want to see the classification accuracy for different "conditions" as a function of time and see how and when the expected/unexpected x repeat/alternate differ in terms of orientation information. However, I think a figure like Figure 5 is also fine provided that all the information is given in the Materials and methods section and the figure caption.

In the Materials and methods section you can also state if you are averaging in time and then computing the classification accuracy or vice versa.

I think the details of which Matlab function they used is unnecessary, but this is key.

[Editors’ note: what now follows is the decision letter after the authors submitted for further consideration.]

Thank you for choosing to send your work entitled "Prediction Error and Repetition Suppression Have Distinct Effects on Neural Representations of Visual Information" for consideration at *eLife*.

Your letter of appeal has been considered by a Senior Editor and a Reviewing editor, and we are prepared to consider a revised submission. However, I should point out that the reviewers have requested a new analysis and the reviewers and editors are all in agreement that our continued consideration of this MS would be contingent on the outcome of this analysis.

I paste below the comments from one of our reviewers (reviewer 1; reviewer 2 was broadly happy with the MS you resubmitted). There are 3 points. We agree that points 1 and 3 can be dealt with by acknowledging limitations to the data/paradigm in the discussion or elsewhere in the main text. Point 2 is more thorny and the reviewer requests a further analysis, because the paper seems to hinge principally on this finding. You have argued that baselining in this period is unbiased because of balancing of trial types in your design. The reviewer is worried (and we agree) that this is not an adequate response, because on expected trials, the baselining will remove trial-specific information that relates to the expectation of the stimulus and potentially affect the levels of encoding of expected vs. unexpected stimuli. To be crystal clear, I precis from our discussion:

"The interval the authors use as the baseline for all their analyses (as far as i can tell), is the interval between gratings 1 and 2, i.e. the time at which they can form an expectation about the upcoming stimulus. So, if there is a neural expectation signal in this interval, using this as baseline affects all (decoding) analyses. For instance, in their Figure 4B, decoding of expected gratings is worse than unexpected – could this be because for the expected trials they've subtracted a pre-stimulus expectation signal that was present in the baseline?"

The reviewer requests that you replicate this analysis baselining from a different period (e.g. before stimulus onset when there is no risk of contamination with expectation signals). Contingent on this analysis working (which, in the event that your explanations were correct, it certainly should), and on satisfactory addressing of points 1 and 3 with discussion of limitations, I think we would be happy to go forward with the paper. I think if an analysis of the sort suggested does not work, the reviewers might be skeptical about whether the claims in the paper stand up.

*Reviewer 1:*

I find the univariate EEG much easier to interpret now, and I applaud the authors for making the effort to improve this. However, I still have concerns about the decoding analyses.

1) With regard to the strong off-diagonal decoding, it seems the authors themselves do not fully understand this either. They suggest it may be to do with the ongoing processing of grating 1 during the processing of grating 2, but would such ongoing late processing really be expected to lead to almost as strong off-diagonal as diagonal decoding, as Figure 7 suggests? I also find the very strong decoding signals in the pre-grating1 testing time disturbing; why would there be decoding prior to presentation of the first grating? (I come back to this below as well.)

2) In response to my concern about the baseline procedure, i.e. baselining on the interval between the two gratings, the authors state that this is not a concern, since the trials were balanced "so there was an equal number of different orientations of the initial grating leading to each subsequent (unexpected) grating within a pair." I don't understand how this addresses my concerns. They say, "the difference between the conditions only arises after presentation of the second stimulus in the pair". While that may be true when comparing an expected repetition to an unexpected alternation, and an expected alternation to an unexpected repetition (i.e., in those comparisons, participants had the same expectation), it is *not* true when comparing e.g. an expected repetition to an unexpected repetition. To be clear, I am not raising a concern about the signal from grating 1 itself carrying over into the grating 2 period, I am concerned about an expectation signal, induced by grating 1 and the context (repetition vs. alternation block), being present in the interval between gratings 1 and 2, i.e. the baseline period.

To be specific, what I am concerned about is that there may be a neural signal in the interval between the two gratings that reflects the expected orientation. This does not need to be carry-over from the bottom-up signal caused by the first grating, it may be a genuine top-down expectation induced template of the expected orientation. If such a pre-second-stimulus expectation template exists, then in the 'Expected repetition' trials the current baselining procedure would lead to confounds, since this template is in the baseline period, and therefore a negative reflection of this template would be introduced throughout the rest of the trial timeline, i.e. during grating 1 and grating 2. This process would affect the different conditions differently, since in 'Unexpected repetition' and 'Expected alternation' trials participants cannot form an expectation of a specific trials, hence no expectation template would be expected to be present, and in the 'Unexpected alternation' trials a template might be present, but it doesn't match the presented orientation (of the second grating). Regardless of this specific confound, in more general terms baselining on a period during which participants form expectations in a subset of the trials seems problematic. The only way I would see of addressing this issue, is to baseline the trials on the EEG data prior to the first grating, when no expectation can be formed yet. (Though from their response to some of the other points, I take the authors as saying that the relationship between the orientation of the second grating on trial n and the first grating on trial n+1 is not perfectly controlled, in which case this approach would not be valid. And indeed, in Figure 7, there seems to be strong decoding signal prior to the presentation of grating 1, which would either be caused by some relationship between grating 2 on the previous trial and grating 1 on the current trial, or by the baselining concern I raise above.)

3) This is a more general point about the experiment design, that I didn't raise before as I only realised it fully in this round. Expected alternation trials were as follows: a certain orientation was presented as the first grating, and then participants could form the expectation that it would alternate, meaning that it was equally likely to be any other orientation. So that's a very unspecific expectation. By contrast, in a previous study orthogonalising expectation and repetition (Todorovic and De Lange), 'expected alternations' meant that participants had a 75% valid expectation of exactly which stimulus would appear. That is, in that study, participants had a 75% chance of predicting the exact stimulus coming up in both expected repetition and expected alternation trials, making those two conditions nicely symmetrical. In the current study however, expected alternation trials ('any orientation can come up, just probably not the one I've just seen') are not at all similar to expected repetition trials, and it seems a bit of a stretch to average these two conditions together and call them 'expected' orientations. In other words, I'm not sure how successful the current design is in orthogonalising expectation and repetition.

---

## [Author Response]

All the reviewers expressed enthusiasm for your work and recognised the quality of the study and the potential import of the findings. Their concerns focused on a number of issues that would need to be addressed for a revised manuscript to be acceptable to eLife.1) The details of several of the analyses were unclear, as noted by all three reviewers. In particular, exactly what was used for training and what for test was unclear for some of the later figures. There is also a seeming discrepancy between Figure 2C and the text. Like reviewer 1, I was surprised by the form of the ERPs in Figure 2; please clarify how these were obtained, if not by standard methods. Reviewer 2 also questions the interpretation of the results presented in Figure 7; it would be important to address that at revision.

We have extensively clarified, throughout the revision, the exact nature of test and training datasets and the backwards decoding analysis. We also provide full details on the statistical tests used throughout the paper, and have clarified the form of the ERPs displayed (see our responses to reviewer 1, comment 2; and reviewer 3, comment 3).

2) The reviewers noted that the Introduction and Discussion section were at times misaligned with the results themselves. In particular, the allusion to predictive coding seemed to be quite loose in places; please make clear which results support predictive coding, and which do not. It would be important for this work to be discussed in the context of previous studies using ERPs/MEG in conjunction with this paradigm, such as Kok et al., 2017. The results described here seem very different from those in that paper (and also discrepant with de Gardelle et al., 2011); the reviewers ask that you at minimum acknowledge and discuss these inconsistencies.

We have revised the terminology used throughout the manuscript and have focused in particular on our use of the term “predictive coding”. We have now better aligned our experimental manipulations and results with the content of the Introduction and Discussion section. In the revision, we have drawn attention to the differences between the findings of Kok et al., (2017) and the results of the current study (see responses to reviewer 1, comment 9).

3) Reviewer 2 raises a key issue about the provenance of the neural signals. Are you able to offer any reassurance that these signals are really "perceptual" in origin?

Forward encoding modelling uses linear regression to find patterns of neural activity selective for stimulus features, in this case orientations. The analysis, therefore, focuses on how repetition and expectation affect stimulus features, which are perceptual features. There is some suggestion that this signal is most apparent in electrodes over the occipital-parietal cortex, but the low spatial resolution of EEG makes us reluctant to draw finer grained anatomical distinctions about the exact cortical loci of the effects we observed; indeed, this was not the central aim of our study. We have extensively revised the Discussion section to clarify the analyses and also to highlight any limitations in the conclusions we can draw (see responses to reviewer 2, comments 1-3).

4) Reviewer 3 asks how the decoding varies with the interaction of the two factors. This seems like a potentially important point that is unaddressed in the current version.

We have included a number of additional analyses in the revised manuscript that test for interactions between repetition suppression and expectation. We used a number of different approaches to examine these effects, but consistently found no significant interaction between the two factors. These analyses support the major conclusion of the current work, namely, that repetition suppression and expectation have distinct effects on sensory representations (see our response to reviewer 3, comment 2).

Reviewer #1:This study by Tang et al., investigates the effects of expectation and repetition on the neural processing of visual stimuli. The authors use both conventional EEG amplitude measurements, as well as forward model decoding to study the neural representations of the stimuli. They find that unexpected stimuli are represented substantially better in the neural signal than expected ones – an intriguing result. This is an interesting topic, investigated using excellent methods that seem executed well. I do have some questions, many of which are for clarification.1) The authors discuss a prediction error (Expected < Unexpected) effect, but from Figure 2C it seems that expected stimuli (blue) evoke more activity than unexpected ones (red). Please explain.

The prediction effect in which the unexpected stimulus evoked a more positive response than an expected stimulus was found over a cluster of central-frontal electrodes. Over the occipital channels, there was more negativity associated with the unexpected stimulus. For Figure 2C, we plotted the effect from a collection of occipital-parietal sensors (O1, O2, Oz, POz, PO7, PO3, PO8, PO4) to facilitate comparison with the repetition suppression effect. If we had used a bandpass filter there would have been a larger negativity for unexpected stimuli (relative to expected stimuli) over these electrodes.

2) I am not an EEG expert, so please excuse me for my naivety, but I would have expected to see ERPs to consist of early components of alternating polarity, i.e. P1-N1-P2-N2-P3. However, here the amplitude of the EEG signal is only ever positive post-stimulus – is this the result of a specific aspect of the paradigm, or the preprocessing of the data?

This relates to the previous question. The shapes of the ERPs in the present study are driven by two main factors: (1) the brief (100 ms) presentation time of the grating stimuli, and (2) the nature of the pre-processing steps. The traditional shape of ERPs comes largely from bandpass filtering the waveform, generally between 2-5 Hz and 20-40 Hz. This filtering yields the familiar positive and negative deflections in the waveform. In the current study, only a 0.5 Hz high-pass filter was applied at the beginning of pre- pass filter because we did not wish to quantify ERP components. Instead we used a contemporary cluster-based permutation approach across electrodes and time (Oostenveld et al., (2011), to determine whether there were differences between conditions for the ERPs. We wanted to keep the data as close to their ‘raw’ form as possible for use in the forward encoding modelling, which was the main focus of the study, so we could be as near to the actual brain activity as possible and avoid potential confounds that filtering can introduce.

3) When studying the effect of repetition on orientation-selective signals, can the authors distinguish whether any modulations would be the result of actual adaptation/repetition suppression, versus lingering activity evoked by the first grating affecting the decoding? Of course, neural adaptation in early sensory regions is the thing the authors are interested in – but imagine that the first grating is still being processed in, for instance, parietal and frontal areas at the time the second grating is presented. Then the decoder trying to decode the second grating would pick up, and be influenced by, these parietal and frontal signals evoked by the first grating, even though they do not reflect 'true' repetition suppression or adaptation. Is this a concern?

This is an excellent question. It is unlikely that any lingering effects of the first stimulus would have contaminated the neural response because all trial types, both repeating and alternating, were used to train the encoding model. The difference between the orientations of the gratings in the alternating pairs were evenly balanced. This should have penalised the model from finding spatial representations for orientations carried by the preceding stimulus, because in the alternating trials these were uncorrelated with the inputs to the model. This issue might have posed a potential problem had we separately trained the model on each condition.

4) In Figure 6, there is a significant off-axis difference in decoding between expected and unexpected trials, which is very interesting. However, I'm having trouble interpreting the fact that this cluster appears above (rather than below, or both below and above) the diagonal. Which axis reflects 'training time' and which 'testing time'?

We have updated the axis labels in Figure 6 to show training and testing times. We found significant off-axis orientation selectivity when we generalised an early training time to a later test time, but not vice versa (training at a later time and testing at an earlier time). We believe that we found this because the EEG response to the grating was stronger and more consistent shortly after the presentation of the grating, leading to more stable orientation encoding, whereas these responses were less correlated at later time points. The stronger, more correlated signal leads to better encoding performance which helps with generalization. This can lead to asymmetric temporal generalization.

5) Figure 7 requires some clarification. In the text, the authors say "Here, we trained the forward encoding model on the orientation of the Gabor that was actually presented (stimulus driven), or on the orientation that was expected based on the first Gabor in the pair (non-stimulus driven)." Which one was true for Figure 7; trained on the presented or expected orientation? Or did this different between quadrants of the matrices? From the text, it seems to me that the decoder is always trained on the T1 orientations – however, if this is true, I don't understand why there is such strong decoder for both unexpected and expected gratings in the off-diagonal quadrants, given that we're only looking at the alternation trials here. That is, given that the T2 orientation different from the T1 orientation, how come it's possible to train on T1 (i.e. negative training times) and decode T2 (positive testing times)? Of course, an argument could be made for the unexpected alternation condition, given that there T1 is expected to repeat, but this is not true for the expected alternation condition.

We apologise for this ambiguity. For this particular analysis we used the orientation of the first Gabor in the pair and compared times when this was expected to repeat (unexpected alternation) and expected to change (expected alternation).

We have rewritten this section (copied below) in the revision to clarify these issues.

“Here, we trained the forward encoding model on the orientation of the first Gabor in the pair to determine how the interaction between the expectation and the incoming sensory information was compared. We anticipated a lower expectation that the orientation would repeat for an expected alternating trial, relative to trials in the unexpected alternation condition, where the repeat should have been expected.”

6) "More interestingly, in the unexpected alternation condition there was significantly better on-axis orientation selectivity for the T1 orientation between 150 and 300 ms after the onset of T2, relative to the expected alternation condition (upper right quadrant of right panel in Figure 7)." So, if I understand correctly, in the post-T2 time window, there is both increased evidence for the T1 orientation (Figure 7), as well as for the T2 orientation (Figure 6), for unexpected compared to expected gratings? This is quite interesting, and surprising. Can the authors confirm that this interpretation of their effects is correct?

Yes, this is our interpretation of this result. We interpret the increased information about both the first and second Gabors as reflecting what the larger MMN response represents. The second Gabor is registered as an error and the prior is updated in light of this error and the previous expectation.

7) Was the decoder always trained on an equal mixture of expected and unexpected and repeating and alternating trials, to make sure the decoder was not biased towards one of these categories? For instance, given that expected gratings are much more numerous than unexpected ones, the decoder may be better at decoding expected trials if the trial counts are not balanced during training.

We trained across all types of trials (expected/unexpected, repeated/alternating, and target trials) before splitting the results by trial type. We did this so we had equal power to detect effects across conditions, and to avoid biasing the results in favour of any particular outcome. Note, for example, that we actually observed better orientation selectivity for unexpected stimuli than for expected stimuli, which is the opposite of what one might predict if the results were being driven purely by trial numbers across conditions. We have clarified the nature of the training and test sets in the revised manuscript (subsection “Forward encoding modelling”).

8) "It is noteworthy, however, that in their study Kok et al., (2012) employed a 'backward' decoding analysis to quantify sharpness, rather than forward encoding as here, which might account for the discrepant findings." I'm not sure I follow this reasoning, given that the authors replicate their own findings using a 'backward' decoding analysis as well. Also note that an MEG study using the same paradigm (Kok et al., 2017) did use forward modelling, and also found improved decoding for expected vs. unexpected gratings.

We agree with this point. We have removed these sentences from the Discussion section of the revised manuscript, and added the reference to recently published Kok et al., (2017) paper.

9) "An intriguing possibility is that combining predictions generated across distinct cortical areas (e.g., visual and auditory) leads to sharpening of tuning, whereas predictions generated within a single cortical area lead to gain modulation." This in indeed an intriguing possibility. I would like to note, however, that it seems unlikely that the predictions in the current study are generated in a single cortical region (e.g. V1), given their complexity. After all, it is not simply grating 1 that predicts the orientation of grating 2, but the orientation of grating 1 in conjunction with the current context ("am I in a repetition or alternation block?"). This type of context seems unlikely to be encoded in early visual cortex but may perhaps be signalled by higher level regions in frontal cortex (or elsewhere) instead.

The reviewer is absolutely correct, and in fact this was the sentiment we wished to convey in the original manuscript. We apologise for the lack of clarity. We have now rewritten the relevant paragraph in the revised manuscript to better explain our interpretation (Discussion section).

10) "The data were then baseline corrected to the average EEG activity from -100 to 0 ms before the presentation of the second Gabor pair." The phrasing is unclear. Does this mean that activity was baseline before presentation of the first or the second member of each pair?

Thank you for drawing this ambiguity to our attention. We have revised the sentence to read: “The data were then baseline corrected to the mean EEG activity from -100 to 0 ms before the presentation of the second Gabor in the pair.”

Reviewer #2:Tang et al., present an EEG study in which they investigate how prediction error and repetition suppression affect neural representations of visual information. They manipulated two factors: (i) whether two stimuli repeated in a trial or not; (ii) whether a repeat was very likely to occur or not. The authors performed a forward encoding analysis to characterize orientation selectivity. They show that repetition suppression decreases overall responsiveness but does not affect orientation selectivity. In contrast, unexpected stimuli, when compared to expected stimuli, only showed a minor difference in overall responsiveness, but instead seemed to be associated with a larger gain in orientation tuning curves without a change in tuning width.This study addresses important questions and provides relevant findings on these issues. The experimental design and implementation is done sufficiently well, the analyses are well thought off, and the manuscript is overall clearly written.Nevertheless, several major revisions will be necessary before the actual implications of the findings are clear. I will first go into a few important concerns.First, it is unclear to me which neural signal underlies the orientation selectivity that is measured. Even if we would accept that it is a signal from visual regions, it must be very far removed from the responses of single neurons or orientation columns. What are the different possibilities, and what are the implications of this uncertainty for our understanding of the neural correlates of prediction error processing? However, before such questions are addressed, it first has to be clarified to what extent the findings relate to visual processing. If I understand correctly, the forward encoding modelling to estimate orientation selectivity considers all electrodes across the entire brain. Therefore, it is impossible to determine which electrodes contribute to the orientation selectivity and its modulation by expectations. Is it possible to perform further analyses to clarify this? Imagine that the expectation effect would be largely driven by frontal electrodes, then the results would not necessarily tell us much about perceptual processing.

As with all non-invasive imaging methods used in humans, including EEG, fMRI and MEG, the neural signals measured are a proxy for a variety of biological signals, including neuronal spiking, local field potentials, etc. Here we chose EEG for its fine temporal resolution as we wished to examine the time course rather than the discrete anatomical locus of repetition and expectation effects in response to simple visual stimuli. As noted by the reviewer, therefore, we need to be cautious in attributing any measured effects to distinct neuronal populations. Only invasive methods employed in animal models can achieve this kind of precision. On the other hand, it is reasonable for us to want to better understand the nature of prediction and repetition effects in people, and indeed there is a rich published history using these non-invasive approaches in humans, much of which we cite in our manuscript.

With regard to our study, we estimated orientation-selectivity using all EEG electrodes across the scalp, consistent with previous pioneering work that has employed this data-driven approach (Garcia et al., 2013; Myers et al., 2015). Figure 3D shows how the model allocates weights to each electrode across subjects at two different time points (50 – 100 ms, and 150 – 200 ms). As is clear from these topographies, the electrodes over occipital and parietal regions are more highly weighted than electrodes over other regions, including frontal areas, suggesting that it is these posterior electrodes that contribute most to the orientation-selective response we measured. This finding is consistent with previously-published work showing orientation selectivity over visual areas in EEG, MEG and fMRI recordings (Cichy et al., 2015; Cichy et al., 2014; Marti and Dehaene, 2017; Stokes et al., 2015).

Having said this, we remain agnostic as to whether more anterior brain areas might also contribute to the effects we found. As noted above, non-invasive imaging measures, including fMRI, provide only a proxy for activity at the level of single neurons. Indeed, the sluggish BOLD response measured with fMRI would not be capable of revealing the fine temporal structure of repetition and expectation effects we have reported here using EEG. It is for future investigations to determine whether individual neurons in visual and non-visual areas might show activity consistent with the whole-brain patterns we have uncovered. To this end, we have recently commenced experiments in which we are measuring neural activity using fMRI in humans and calcium imaging in mice, with a view to determining the effects of repetition and expectation in early visual areas. Preliminary results suggest that prediction effects do indeed arise in primary visual cortex.

We have added a paragraph in the revision discussion to clarify these issues (Discussion section).

Second, and partially related to the first, subsection “Repetition suppression and prediction error affect the overall level of neural activity” and Figure 2 show that repetition and expectation effects are found both in occipital-parietal and in frontal electrodes. The effect for repetitions in occipital-parietal electrodes (repeat < alternation) is in the expected direction. However, the expectation effect seems to be in the wrong direction in occipital-parietal electrodes (expected > unexpected). From a predictive coding perspective, unexpected repetitions/alternations should result in a higher prediction error and thus higher response. Just like alternation trials result in a higher response. How can we interpret the further multivariate results if this main effect of expectation is hard to reconcile with what we see for repetition suppression and with what we would expect based on e.g. the fMRI experiment of Summerfield et al.,?

Please see our answer to this question in response to reviewer 1, comment 1.

Third, and again somewhat related to the previous concern, at many points the relationship between prediction errors, repetition suppression, and neuronal adaptation is not characterized well. A few examples: In the Abstract prediction errors are claimed to provide a possible explanation of repetition suppression. However, in previous work (as well as in this study), expectation effects are dissociated from repetition suppression. The latter can be explained perfectly well by stimulus driven adaptation effects. These expectation effects are smaller and less robust than stimulus driven adaptation, suggesting a minor modulatory role of perceptual expectation.

The reviewer is correct that our study, as well as previous work, has suggested that expectation and repetition suppression are separable. However, to our knowledge no previous study has examined how these factors influence the representation of sensory information (as opposed to overall changes in response amplitude). It is important to test this idea, as a strong version of predictive coding theory argues that expectation can fully explain repetition suppression. From this standpoint, adaptation is explainable by expectation; the system expects to see a repeat, and when it occurs the neuronal response is reduced. We believe it is important to test whether, and how, expectations modulate adaptation. We have revised the Abstract, Introduction and Discussion section to clarify how the current study addresses this issue.

Introduction: "Here we asked whether a predictive coding theory can account for the changes in neural representations observed with repetition suppression." How is this question answered?

The main conclusion of our study is that prediction error, but not repetition suppression, affects the neural representation of basic perceptual information (in this case, the gain associated with orientation tuning). This suggests that they are indeed separable processes. We have clarified this issue in the revision (Discussion section).

Fourth, the authors should adjust their claims and interpretation to the strength of the evidence provided, from abstract to Discussion section. First, in order to do this, the statistical reporting should be more specific. The authors should report actual p-values (of course rounded to a reasonable precision), instead of arbitrary thresholds, such as p<.025 or p<.05. Also, for the statistics shown mostly in figures, such as Figure 6 and Figure 7, it would be important to mention the highest p-value obtained, and not just say p<0.05. Most effects seem to be close to 0.05, which should be reflected upon. One p-value signals a very low probability (p =.0049; subsection “Prediction errors increase the amount of orientation-selective information contained within patterns of EEG activity”), however, it is the result of a circular analysis. The time window on which this analyses was done was pre-selected based on a statistical test on the same data. This should be indicated. Overall, the amount of evidence presented in this paper seems sufficient for a publication but not strong enough to just pass without further discussion and adjusting the interpretations to the strength of the evidence.

We now have included exact p-values for all reported statistical tests.

In the revised manuscript, we have provided a more thorough description of the analysis reported in subsection “Prediction errors increase the amount of orientation-selective information contained within patterns of EEG activity””. Briefly, we initially tested all four parameters of the Gaussian fits to the data (amplitude/gain, centre orientation, width/tuning, and baseline) using non-permutation cluster-based testing (Oostenveld et al., 2011; Wolff et al., 2017), which is a conservative analytic approach. This yielded a difference between expected and unexpected stimuli at an early time point for the amplitude (gain) parameter, but not for the width parameter. We averaged the data across the significant time points to increase signal-to-noise, then applied a conventional t-test to each of the four parameters again to ensure we did not miss any small but significant effects. Given that we had already found a significant main effect of amplitude (gain) in the initial (conservative) cluster-based test, the conclusions we draw from the subsequent t tests do not alter the central conclusion of a change in gain with expectation.

Fifth, the orientation selectivity is determined and tested using the second gabor in the trials. If you find a difference between two conditions A and B, with A showing better tuning than B, how can you know whether it is A that has an increased tuning or B a decreased tuning? To me it seems that the orientation selectivity on the first gabor in the trials would be a good baseline/reference. So why did you not include the first gabor in the trials in your orientation selectivity analyses? You conclude "prediction error was associated with a significantly increased orientation-selective response through a gain modulation soon after the stimulus was presented", but how can you differentiate this from "The absence of prediction error was associated with a significantly decreased orientation-selective response through a gain modulation soon after the stimulus was presented". When split in subsets of trials according to the relationship with the second stimulus, these analyses on the first gabor would also serve as a useful statistical control for the analyses presented in Figure 2. If the statistical thresholding is stringent enough, then doing the same analysis on the first stimulus should not give false positives.

Using the first target as the comparison stimulus would be misleading in this context, for the following reason. We inserted a randomised inter-trial interval of 600 – 1250 ms (i.e., between the second stimulus in one pair and the first stimulus in the next pair), but there was a fixed interval (500 ms) between the first and second stimuli within each pair. This design feature means that there was less consistent ongoing activity for the first grating compared with the consistent activity before the second grating. This difference is likely to make the suggested comparison problematic. The issue is avoided by comparing how the representation of the second grating is affected by the different stimulus manipulations which have matched properties. We have now clarified throughout the revision that our finding of increased orientation selectivity is relative to the expected stimuli.

Reviewer #3:I found the paper very interesting.While they show that the univariate effects of repetition and expectation are comparable (same areas, different sign), they show that while repetition doesn't affect the selectivity of representations, expectation leads to an increase in selectivity by gain modulation.Given the importance of the multivariate analysis in directly leading to the main claim of the paper, I have a few comments on it:1) The authors use an encoding approach to test for information processing. That is very clearly explained in the Materials and methods section. At the last step, they fit a Gaussian (Equation 4), which has 4 parameters. So, for example Figure 3C quantifies the orientation selectivity by parameter A. Now whilst that is an interesting fact, it might be the case that there are other differences in the two (e.g. that σ is smaller in the repeated than the non-repeated).

This is an excellent point. We have now included this analysis in the revised manuscript (subsection “Prediction errors increase the amount of orientation-selective information contained within patterns of EEG activity”). Our original conclusions remain unchanged.

2) The multivariate analysis that compares orientation selectivity in repeat/ alternate and expected/unexpected is key to the claims of the paper. However, the authors could have looked at all 4 possible combinations (similar to what they do for the LDA visualized in Figure 5). For example, it would be interesting to test if orientation selectivity in the expected is different when the stimulus is repeated or alternated. Figure 4 would be a good summary if there was no interaction between the two at any time point.

Thank you for this suggestion. We had in fact already performed precisely your suggested analyses but chose not to include the details in the original manuscript. We have now included these further results (subsection “Prediction errors increase the amount of orientation-selective information contained within patterns of EEG activity”), as requested. To summarise, using a number of different approaches we found no evidence for a significant interaction between the factors at any time point.

3) Figure 5 is an interesting control analysis. However, there is no information on how they performed cross validation, what time points are averaged to give the result and how stable the results are!

We apologise for omitting this information. We have now included a detailed description of the methods in the revised manuscript (see subsection “Multivariate pattern analysis”).

4) Although I find the cross-temporal analysis on the weights from the forward encoding model interesting, I think that some speculations could be directly tested with pattern similarity analysis. For example, for the unexpected stimulus, they could directly test whether the representation of the unexpected is reactivated at 300-400 ms. They could do non-parametric statistics for the inference. I also make the same suggestion for analysis of Figure 7.

We thank the reviewer for making this suggestion, as we are not experts in RSA. We conducted the analysis in Figure 6 and Figure 7 to determine whether the same pattern of neural response was stable over time. Consistent with previous work, we found that selectivity was best on-axis, but that there was still significant generalization of these classifiers. We found that generalization is enhanced when there is an unexpected stimulus, and because this pattern of activity was orientation selective, we concluded that the same pattern of neural activity was re-activated. We are hesitant to introduce yet another analysis approach in the current manuscript, because it would require a different approach to training the model, and there are already a large number of distinct analyses in the current study.

[Editors’ note: the author responses to the second round of peer review follow.]

Our decision has been reached after extensive consultation between the reviewers. Based on these discussions and the individual reviews below, we regret to inform you that we have decided to reject your submission in its current form.As you can see from the comments below, all of the reviewers – and in particular #1 and #2 – felt that their comments had been insufficiently addressed at revision. The reviewing editor has read your revised manuscript and rebuttal with care and agrees. He has flagged the following major issues major issues are as follows:(i) One major premise of the manuscript is that you claim to see a univariate effect of unexpected > expected trials. However, the plot you present in Figure 2 shows precisely the converse. The reviewers found your argument that this effect reverses over more anterior sites (not shown) unconvincing, particularly given that the major focus of the paper is on predictive coding in hierarchical sensory systems. This, coupled with the unconventional presentation of the ERPs (which do not resemble potentials that the reader would typically expect to see over visual regions), led the reviewers to doubt the claims made with regard to this signal.

We originally presented event-related potentials (ERPs) that had not been bandpass filtered, in contrast to the convention for showing mismatch negativity waveforms. This caused the ERPs to have a positive polarity over occipital channels. We took this approach because bandpass filtering can introduce artefacts, and because such filtering is not necessary when peak analyses are not being undertaken. However, to address the concerns of reviewers 1 and 2, we now show the ERP results in the conventional manner. The ERPs were bandpass filtered so that they produced the stereotypic peaks and troughs, as shown in the revised Figure 2. As expected, a greater negativity was found for unexpected than for expected stimuli (Figure 2C) over occipital-parietal electrodes. The ERPs we show are thus entirely consistent with those reported previously for the visual MMN (Näätänen et al., 2007; Saarinen et al., 1992).

(ii) Reviewers #1 and #2 raised substantive issues about the train/test approach that you took for the multivariate analyses. The paradigm you have employed allows the viewer to form expectations about the second grating on the basis of the first. This means that the neural signal elicited by the first grating may partly encode an expectation of the second grating, and the neural signal elicited by the second grating might partly include carryover from the first grating. Uncertainty over whether the effects you described have a straightforward interpretation was exacerbated by the asymmetry in the cross-validation plots. The reviewers remain unconvinced by your rather cursory replies to their questions regarding this point.

As we highlight in detail in our response to reviewer 1 (Comment 4), we employed a balanced number of orientations for the initial grating within each pair. The orientations of these initial gratings were not only uncorrelated over trials, they were also balanced for every orientation of the (unexpected) second grating. Critically, we used a regression-based method to recover orientation selectivity, so that the effects of the first stimulus on the second would sum to zero across trials, and thus could not bias the results for encoding of the second stimulus within each pair.

Nevertheless, despite these points in favour of the analytic approach we used, we have now also conducted the exact analysis suggested by reviewer 2 (Comment 4). This revealed the pattern of results expected by the reviewer (as outlined in detail below) but suffers from a stimulus confound as we argued previously.

Having undertaken this analysis as requested, we again wish to point out that the paradigm was deliberately designed such that the orientation leading into the second grating was precisely matched across conditions. This allowed us to directly compare the conditions of interest without any stimulus confounds. By contrast, it is important to note that the first grating within each pair was subject to different conditions across the trials. For example, the interval between the offset of the second grating in the preceding trial and the onset of the first grating in the next trial was selected randomly between 650 and 750 ms. Thus, by definition, the appearance of the first grating within each pair was inherently unpredictable (because the timing of its onset was randomised across trials). Moreover, no effort was made to control the orientation of the stimulus preceding the first grating in each pair. We therefore stand by our contention that the results of our original analysis are unconfounded, whereas those arising from the suggested additional analysis are. If the reviewers and editors feel it is imperative we report this additional analysis in the revised manuscript we shall do so, but at this point we have not included it.

(iii) The reviewers still feel that the link to predictive coding is very loose. The manuscript continues to be motivated by the predictive coding framework, even though several of your results seem to directly contradict what might be expected under this theory. The reviewers found the exegesis to be rather contradictory, and the theoretical claims confusing in the light of the results.

As outlined in detail below, we have endeavoured to provide a more nuanced account of predictive coding theory and the hypotheses that arise from it in terms of feature encoding, as addressed in our manuscript. In particular, we highlight that predictive coding theory has been largely based on analyses of the overall magnitude of neural responses to unexpected stimuli in human neuroimaging investigations. Critically, previous studies have not determined how prediction and repetition affect the quality of neural representations of elementary stimulus features such as orientation, as we have done. Our data are consistent with some of the predictions arising from the theory, and are inconsistent with others. We have now highlighted these inconsistencies in the revised manuscript and have drawn attention to some of the constraints our data impose on the theory. Note, however, that our aim in framing our work in terms of predictive coding should not be construed as endorsement of the theory. We remain agnostic as to whether predictive coding theory provides an optimal conceptualisation of the effects of repetition and expectation on neural responses and believe the results we report add value to the literature irrespective of this influential framework.

(iv) There are continued concerns from reviewers #2 and #3 that your analyses are at least partly circular, and that there may be oddities introduced by your baselining techniques (reviewer #1).

We deliberately took a conservative approach in our statistical analyses, and were careful to avoid circularity. As outlined above, we have now conducted the analysis suggested by reviewer 2. As for reviewer 3, we have implemented further checks to avoid circularity. We selected the peak classification accuracy for each participant across a wide timing window, and this window was identical between conditions (Repeat/Alternate and Expected/Unexpected). We did not select time points based on differences between the conditions, but instead found the peak classification accuracy across the trial. We have included full details of our statistical choices in the Results section and Materials and methods sections. In response to reviewer 1, we have made explicit that an equal number of orientations went into every prediction error, and these were uncorrelated across trials for each observer. As we employed a regression-based forward encoding approach, any influence of the first and second gratings will cancel to zero and thus not systematically bias responses to the second grating.

Together, this prompted the decision to reject the manuscript as it stands. If you would like to appeal this decision, you are welcome to do so, but I would suggest that it is unlikely that the reviewers will be sympathetic to an appeal without a serious revision to the paper that addresses their points in full. For my part, I think that this manuscript has promise, but I feel that there are simply too many inconsistencies in the current version to warrant publication in eLife.Reviewer #1:Unfortunately, I feel that not all concerns have been satisfactorily addressed. In its current form, I am not fully confident that the results described reflect effects of expectation on perceptual representations. The authors' responses to my concerns expressed in the previous round have not fully taken away these concerns, as I detail below.1) Original point 1. I'm still not quite sure how to interpret the amplitude results displayed in Figure 2C-D. It seems to me, from Figure 2C, that expected stimuli evoke a larger response than unexpected ones, yet the authors interpret this as a prediction error effect, with a larger neural response to unexpected stimuli (over frontal electrodes). In their response, they state that there was "more negativity associated with the unexpected stimulus", but I'm not sure how this follows from Figure 2C-D.2) Original point 2. According to the authors, the fact that the ERP waveforms is (almost) only positive, rather than a sequence of positive and negative components, is the result of the minimal preprocessing applied (only a 0.5 Hz high-pass filter). However, from my limited experience with EEG, I would expect positive and negative components even with minimal preprocessing. I'm still somewhat confused by this.

Points 1 and 2 are closely related, and so we have chosen to deal with them together in our response. We thank the reviewer for highlighting this ambiguity in our reporting of the original ERPs. We have conducted a new analysis that brings our results in line with the mismatch negativity literature, and which should therefore alleviate these concerns. To briefly reiterate our earlier explanation of the original result, the waveform shape is largely driven by the filtering used and the presented stimulus. Traditionally, ERPs have been band-pass filtered so that simple peak analyses can be applied. We believe this approach is outdated, and that modern analytic approaches are sufficient to reveal the actual, rather than processed, brain response (and that these approaches are also less susceptible to experimenter degrees of freedom). However, we acknowledge these subtle issues aren’t obvious to those outside the field, and we have therefore included bandpass filtered results and conducted a more traditional peak analysis, so the results adhere better to the familiar pattern. This analysis showed a larger positivity (P1 component) to the alternate than the repeating stimuli. Critically, there was a significantly greater negativity (N1 component) for unexpected stimuli than for expected stimuli, in line with the widely reported visual mismatch negativity (Garrido, Sahani and Dolan, 2013; Näätänen et al., 2007; Saarinen et al., 1992).

Note that this analysis merely provides confirmation that the expectation manipulation elicited a surprise response in brain. It does not address our central question of how visual features – in this case orientation – are affected by expectancy. It is this aspect of predictive coding theory that we go on to investigate using multivariate forward encoding analyses.

3) Original point 5. I still don't understand why there is such strong off-diagonal decoding in Figure 7, given that the orientation of the second grating was randomized, as the authors state in the legend. In the orientation of the second grating was randomized, how is it possible to decode it when training on the first grating?

We believe the strong off-diagonal encoding of the first grating during presentation of the second grating reflects ongoing processing of the initial stimulus. It is possible to see prolonged encoding of the first grating even after presentation of the second because the orientations of the two gratings within each pair were uncorrelated. One interpretation of this effect is that it reflects an ongoing comparison of the orientations of the first and second stimuli, but we acknowledge that at present we do not have a definitive explanation for the off-diagonal encoding observed. We hope our findings will provoke readers to generate and test their own hypotheses with respect to this interesting and reliable effect.

4) In response to my original point #10, the authors explain that the EEG data were baseline corrected to the interval -100 to 0 ms before presentation of the second Gabor in the pair. Isn't this problematic, in a paradigm in which grating 1 can lead to a prediction about grating 2? If there is an expectation signal present just before the appearance of grating 2 (as is quite likely), this would 'contaminate' the baseline, and effects found elsewhere (before -100 ms or after 0 ms) might actually reflect a negative image of expectation effects present in the pre-stimulus 2 interval.

We carefully balanced the trials so there was an equal number of different orientations of the initial grating leading to each subsequent (unexpected) grating within a pair. For repeating blocks, the expectation was that the orientation would repeat; thus, the unexpected stimulus was any orientation apart from the expected one. Critically, the difference between the conditions only arises after presentation of the second stimulus in the pair, once again ensuring the baseline period is matched across conditions. As we trained across all trial types, the expectations contained within the baseline period averaged to zero, removing any potential for contamination. We have included a discussion of these issues in subsection “EEG acquisition and pre-processing” of the revised manuscript.

Reviewer #2:I thank the authors for their revision, which addresses some of the earlier concerns.There are a few remaining issues, addressing them would improve the manuscript further.About the localization of the signal:Central to predictive coding theories is the idea that sensory predictions which are generated higher up in the hierarchy modulate responses in lower levels of sensory cortices. This statement is much more specific than the more general idea that: recent experience establishes expectations in the brain, which manifest somewhere and somehow in the encoding of sensory input. In the current state of the manuscript, this important nuance seems to be missing. Given that predictive coding theories have very specific hierarchical predictions, the argument that the authors wanted to "limit experimenter degrees of freedom potentially introduced through the post-hoc selection of subsets of electrodes" is simply not valid here. This is not a trivial issue, because as long as we do not constrain a theory, it cannot be falsified.I understand that the authors might not want to or might not be able to localize their effects of expectation, but that limits the extent to which these findings are instructive about predictive coding theory and more generally about how the brain processes unexpected sensory events. Even if the general orientation selectivity is mainly determined by the signal at occipital and parietal electrodes, there is no guarantee that any interaction between expectation and orientation selectivity originates from the same signal in these electrodes. For example, I wonder whether a potential explanation for the discrepancy with Kok et al.,'s work could be that their effects were localized to visual cortex? Perhaps in this study their expectation effects do not originate from in visual cortex, but for example only in frontal areas?Without such information, the findings of this study reduce to: (a) that repetition suppression and expectation effects are dissociable (which already has been demonstrated by for example Todorovic and de Lange, 2012) and (b) expectation increases selectivity for (Gabor) stimuli somewhere in the brain for this particular task. The fact that this would be hard to reconcile with other findings in early visual cortex is then explained away by the argument that "as this is a relatively complex phenomenon it is likely to yield different outcomes when different levels of prediction are manipulated". This reasoning does not echo the idea of predictive coding as universal principle of cortical responses. A good theory should be able to make general predictions that are valid in a (relatively) broad range of situations, without requiring ad hoc explanations for different experiments.The authors should be more explicit about these problems.

We agree with the reviewer that a good theory should make testable predictions. The current work aimed to provide a strong test of one aspect of predictive coding theory, namely, how expectations affect the neural representation of visual feature information. We believe it is important to test this key aspect of the theory because most evidence for predictive coding does not address this issue. As the reviewer correctly recognises, aspects of our results may not fit neatly with the current version of the theory as formulated by Rao and Ballard, (1999) or Friston, (2005). Instead, they provide an important constraint on future versions of the theory.

We also agree that the hierarchal aspect of the theory can only be tested by simultaneously measuring neuronal activity at multiple stages of the processing hierarchy. The requisite temporal and spatial resolution for such a test is likely only possible using invasive recordings across cortical areas in animal models. Here we have introduced a paradigm and data analytic approach which could by readily implemented in animal experiments. We chose to use EEG in our study because it is the most widely-used index of the mismatch negativity effect, this allowing us to compare our results with those reported across this large and longstanding literature.

We have substantially revised the Introduction, Results section and Discussion section of the manuscript to indicate more clearly where the results support (or do not support) predictive coding theory. We have also identified limitations of the current work and provided a more detailed consideration of how invasive recordings in animal models could further test the theory.

Finally, we would like to repeat a point we made in response to the Editors’ comments (above). Our aim in framing our work in terms of predictive coding should not be construed as endorsement of the theory. We remain agnostic as to whether predictive coding theory provides an optimal conceptualisation of the effects of repetition and expectation on neural responses and believe the results we report in the paper add value to the field quite independently of this influential framework.

Specific points:"One mechanism by which the brain reduces its information processing load is to encode successive presentations of the same stimulus in a more efficient form, a process known as neural adaptation": The authors have added references for this sentence, but I think they still can't justify the statement. Right now, the sentence reads like the function of adaptation is a settled case. Given our current lack of understanding of the mechanisms underlying adaptation, any suggestions about its function remain highly speculative. I suggest you change this sentence to "Neural adaptation is one mechanism by which the brain might reduce its information…".

We have changed this sentence as suggested.

In their rebuttal the authors confirm that they find Expected > Unexpected in occipital-parietal electrodes (while Expected < Unexpected in central frontal electrodes). This is a finding that contradicts the hallmark of predictive coding theory, namely that neural response strength signals prediction error. The authors should address this discrepancy. On the other hand, the authors still use the term "prediction errors" in reference to their results. For example, in the Discussion section they say: "We found that prediction errors were associated with increased gain of stimulus representations…". Yet, as they acknowledge, they can't know where the change in stimulus presentations took place, while they only find prediction errors in a frontal cluster.

This same issue was raised by reviewer 1 (comments 1 and 2), and we refer the reviewer to our response to this point above.

Comment 5:I understand that care should be taken for these comparisons, but still it is very valid to compare with the first stimulus. Take the field of single-unit animal studies of repetition suppression, would they not always show data/analyses for the first stimulus also? I think it is an important statistical control for various effects (as I mentioned in my original email).

We have now conducted this analysis as requested, noting the caveats with respect to stimulus confounds outlined in our response to the Editors (see their Point ii). To do this, we had to modify our analytic approach because the orientations presented in the first and second gratings were often not the same. We separated each trial into first- and second-grating epochs, and trained and tested the forward model on the orientations within each epoch. For each condition (i.e., Repeat, Alternate, Expected and Unexpected) we compared the response for the first and second gratings. The Author response image 1 shows the early (79 -185 ms) orientation-selective response to the grating for each condition. As the figure shows, the only marginally reliable difference in orientation selectivity emerged for the unexpected condition, such that the second grating elicited a larger (*p* =. 056) response than the first grating. None of the other comparisons were significant (*p* >. 2). As noted in our response to the Editors (their point ii), we have chosen not to include these results in the revised manuscript as they involve an inherent stimulus confound. If the reviewers and editors feel it is imperative we report the new results, however, we shall do so.

Reviewer #3:I think the revised document is much more clear and transparent.After reading the responses and details of Figure 5, I have a slight concern that it might suffer from some circularity.The figure is very clear and the results are consistent. However, the time period for which these results are displayed can affect the trend.Ideally one would want to see the classification accuracy for different "conditions" as a function of time and see how and when the expected/unexpected x repeat/alternate differ in terms of orientation information. However, I think a figure like Figure 5 is also fine provided that all the information is given in the Materials and methods section and the figure caption.In the Materials and methods section you can also state if you are averaging in time and then computing the classification accuracy or vice versa.I think the details of which Matlab function they used is unnecessary, but this is key.

Our aim was to make the analysis of the data shown in Figure 5 as transparent and straightforward as possible. To clarify our approach, the values used in the analysis were not chosen arbitrarily in terms of timing, but rather were determined by the data themselves. The value selected was the maximum classification accuracy from a wide time window (from 0-600 ms after stimulus presentation) and was exactly the same between all conditions for each participant. These values were then compared between the conditions of interest. Our goal in taking this approach was to accommodate the possibility of inter-individual differences in peak classification across participants without biasing the result to any condition. We also used the maximum classification accuracy to remove the time dimension so all four conditions could be simultaneously visualised and compared. We have now included a detailed description of how we derived the measure in the Results section and Materials and methods sections.

[Editors’ note: the author responses to the re-review follow.]

Your letter of appeal has been considered by a Senior Editor and a Reviewing editor, and we are prepared to consider a revised submission. However, I should point out that the reviewers have requested a new analysis and the reviewers and editors are all in agreement that our continued consideration of this MS would be contingent on the outcome of this analysis.I paste below the comments from one of our reviewers (reviewer 1; reviewer 2 was broadly happy with the MS you resubmitted). There are 3 points. We agree that points 1 and 3 can be dealt with by acknowledging limitations to the data/paradigm in the discussion or elsewhere in the main text. Point 2 is more thorny and the reviewer requests a further analysis, because the paper seems to hinge principally on this finding. You have argued that baselining in this period is unbiased because of balancing of trial types in your design. The reviewer is worried (and we agree) that this is not an adequate response, because on expected trials, the baselining will remove trial-specific information that relates to the expectation of the stimulus and potentially affect the levels of encoding of expected vs. unexpected stimuli. To be crystal clear, I precis from our discussion:"The interval the authors use as the baseline for all their analyses (as far as i can tell), is the interval between gratings 1 and 2, i.e. the time at which they can form an expectation about the upcoming stimulus. So, if there is a neural expectation signal in this interval, using this as baseline affects all (decoding) analyses. For instance, in their Figure 4B, decoding of expected gratings is worse than unexpected – could this be because for the expected trials they've subtracted a pre-stimulus expectation signal that was present in the baseline?"The reviewer requests that you replicate this analysis baselining from a different period (e.g. before stimulus onset when there is no risk of contamination with expectation signals). Contingent on this analysis working (which, in the event that your explanations were correct, it certainly should), and on satisfactory addressing of points 1 and 3 with discussion of limitations, I think we would be happy to go forward with the paper. I think if an analysis of the sort suggested does not work, the reviewers might be skeptical about whether the claims in the paper stand up.

We thank the editors for this clearly worded summary of the residual issue concerning our selection of an appropriate baseline epoch in the reported decoding analyses. We have now undertaken the exact analysis requested by reviewer 1 and replicated the pattern that emerged from our original analysis. We have provided a full explanation of this new analysis in our detailed response to reviewer 1, below.

Reviewer 1:I find the univariate EEG much easier to interpret now, and I applaud the authors for making the effort to improve this. However, I still have concerns about the decoding analyses.1) With regard to the strong off-diagonal decoding, it seems the authors themselves do not fully understand this either. They suggest it may be to do with the ongoing processing of grating 1 during the processing of grating 2, but would such ongoing late processing really be expected to lead to almost as strong off-diagonal as diagonal decoding, as Figure 7 suggests? I also find the very strong decoding signals in the pre-grating1 testing time disturbing; why would there be decoding prior to presentation of the first grating? (I come back to this below as well.)

Thank you for raising this issue. As indicated by the reviewer, interpretation of the off-diagonal decoding result is not straightforward and lends itself to several plausible interpretations that are difficult to disentangle from one another. Consequently, to improve the clarity of the paper, we have removed this analysis entirely. Importantly, none of the main conclusions of our work rely on the outcome of this analysis, and the central message of the paper remains unchanged.

2) In response to my concern about the baseline procedure, i.e. baselining on the interval between the two gratings, the authors state that this is not a concern, since the trials were balanced "so there was an equal number of different orientations of the initial grating leading to each subsequent (unexpected) grating within a pair." I don't understand how this addresses my concerns. They say, "the difference between the conditions only arises after presentation of the second stimulus in the pair". While that may be true when comparing an expected repetition to an unexpected alternation, and an expected alternation to an unexpected repetition (i.e., in those comparisons, participants had the same expectation), it is *not* true when comparing e.g. an expected repetition to an unexpected repetition. To be clear, I am not raising a concern about the signal from grating 1 itself carrying over into the grating 2 period, I am concerned about an expectation signal, induced by grating 1 and the context (repetition vs. alternation block), being present in the interval between gratings 1 and 2, i.e. the baseline period.To be specific, what I am concerned about is that there may be a neural signal in the interval between the two gratings that reflects the expected orientation. This does not need to be carry-over from the bottom-up signal caused by the first grating, it may be a genuine top-down expectation induced template of the expected orientation. If such a pre-second-stimulus expectation template exists, then in the 'Expected repetition' trials the current baselining procedure would lead to confounds, since this template is in the baseline period, and therefore a negative reflection of this template would be introduced throughout the rest of the trial timeline, i.e. during grating 1 and grating 2. This process would affect the different conditions differently, since in 'Unexpected repetition' and 'Expected alternation' trials participants cannot form an expectation of a specific trials, hence no expectation template would be expected to be present, and in the 'Unexpected alternation' trials a template might be present, but it doesn't match the presented orientation (of the second grating). Regardless of this specific confound, in more general terms baselining on a period during which participants form expectations in a subset of the trials seems problematic. The only way I would see of addressing this issue, is to baseline the trials on the EEG data prior to the first grating, when no expectation can be formed yet. (Though from their response to some of the other points, I take the authors as saying that the relationship between the orientation of the second grating on trial n and the first grating on trial n+1 is not perfectly controlled, in which case this approach would not be valid. And indeed, in Figure 7, there seems to be strong decoding signal prior to the presentation of grating 1, which would either be caused by some relationship between grating 2 on the previous trial and grating 1 on the current trial, or by the baselining concern I raise above.)

We understand the nature of the reviewer’s concern here. To address this issue, we employed a new baseline period that extended from -100 to 0 ms before the appearance of the first Gabor in a pair, as suggested by the reviewer, and recomputed the forward encoding model. We averaged the responses over the same early time interval over which the original analysis showed increased orientation selectivity. This confirmed the original pattern of results, such that orientation selectivity was significantly greater for unexpected versus expected stimuli (see Figure 4—figure supplement 1). Again, there was no effect on the width of the representation with expectation. It should also be noted that our original procedure of taking a baseline before the second Gabor is in line with previous work that used variants of the same paradigm (Todorovic and de Lange, 2012). We have included the results of this new analysis from subsection “Expectations increase orientation-selective information contained within patterns of EEG activity” of the revised manuscript (below).

“It might be argued that the particular baseline period we chose for the encoding analyses – namely from -100 to 0 ms before the onset of the second Gabor in each pair – biased the results by incorporating a purely top-down expectation template triggered by the orientation of the first Gabor (Kok et al., 2017). To rule out this possibility, we performed a further forward encoding analysis where we baselined the raw EEG data to the mean activity from -100 to 0 ms before the *first* Gabor in each pair. Critically, this control analysis involved a baseline period over which it was not possible to form a top-down expectation of the orientation of the second Gabor based on the orientation of the first. This analysis yielded the same pattern of results as the original analysis (Figure 4—figure supplement 1), such that the unexpected stimulus evoked significantly greater orientation selectivity than the expected stimulus (*p* =.02). Also in line with the original analyses, the width of the representation was not affected by expectation (*p* =.44), and there was no effect of repetition suppression on orientation selectivity (p =.64). We can thus be confident that the effect of expectation on orientation selectivity that we report here, based on our forward encoding analyses, is not an artefact of the baselining procedure.”

3) This is a more general point about the experiment design, that I didn't raise before as I only realised it fully in this round. Expected alternation trials were as follows: a certain orientation was presented as the first grating, and then participants could form the expectation that it would alternate, meaning that it was equally likely to be any other orientation. So that's a very unspecific expectation. By contrast, in a previous study orthogonalising expectation and repetition (Todorovic and De Lange), 'expected alternations' meant that participants had a 75% valid expectation of exactly which stimulus would appear. That is, in that study, participants had a 75% chance of predicting the exact stimulus coming up in both expected repetition and expected alternation trials, making those two conditions nicely symmetrical. In the current study however, expected alternation trials ('any orientation can come up, just probably not the one I've just seen') are not at all similar to expected repetition trials, and it seems a bit of a stretch to average these two conditions together and call them 'expected' orientations. In other words, I'm not sure how successful the current design is in orthogonalising expectation and repetition.

Thank you for highlighting this subtlety in the design, which we have now clarified in the revised manuscript. To be clear, our design was based on the original Summerfield et al. (2008) work, with Gabor orientation replacing faces while maintaining the same relationships in the alternating trials. Furthermore, the current design is consistent with later work examining the interaction between repetition suppression and expectation (Kaliukhovich and Vogel, 2010; Kovács, et al., 2013).

To briefly explain our choice of design, in the Todorovic and de Lange (2012) study only two or three auditory frequencies were used. This made it easier to present alternating stimuli with a predictable identity. By contrast, in our visual paradigm we used nine different grating orientations to ensure we could measure orientation selectivity at a reasonable level of resolution. It simply was not feasible to have every orientation uniquely predict another orientation in the alternating blocks (e.g., having a 0 degree Gabor predicting a 90 degree Gabor in one block, a zero degree Gabor predicting a 40 degree Gabor in another block, and so on for all possible combinations of stimuli). This would have led to a factorial explosion of conditions. Worse still, such an approach, even if feasible, could have complicated interpretation of the results, as the amount of visual adaptation is itself highly orientation dependent (Blackmore and Campbell, 1969; Zavitz et al., 2016). The rule in the current study was that the orientation of the first Gabor was repeated or alternated for the second Gabor. This rule can be equally violated for repeating and alternating orientations, meaning the conditions are still matched for expectation and repetition. We have now added a consideration of this issue, as requested, from subsection “Experimental Design” of the revised manuscript (below).

“The relationship between the pairs of orientations for the different expectation conditions was based on the original study (Summerfield et al., 2008), and on other studies (Kaliukhovich & Vogels, 2010; Kovács et al., 2013) that examined the interaction between repetition suppression and expectation. In the repeating condition, the orientation of the second Gabor is expected to be the same as the orientation of the first, whereas in the alternating condition the orientation of the second Gabor is expected to be *different* from that of the first. This relationship between the expected orientations of the stimuli in the alternating condition is slightly different to another modification of the paradigm which employed a more limited range of stimuli (Todorovic et al., 2011; Todorovic and de Lange, 2012). Specifically, the paradigm introduced by Todorovic and colleagues used two or three auditory tones of different frequencies. In the alternating condition, the expectation is that one tone will follow another (i.e. 1000 Hz and then 1032 Hz), then this is violated when a 1000 Hz tone is repeated. In this paradigm, an exact frequency is expected in the alternating condition, a design feature that differs from the paradigm used in the current work where there is no specific expectation of the orientation of the second Gabor based on the orientation of the first in the alternating condition. Instead the expectation in the alternating condition is that the orientation will change, and this can be violated by repeating the orientation. In this sense, there is no specific expectation about the second orientation in the alternating condition. Instead, the rule is about alternating or repeating the first orientation. We did not implement the Todorovic et al. paradigm because the combinatorial explosion of stimulus conditions needed to measure orientation selectivity (such that every orientation is predicted by another orientation). Future work could investigate how this subtle change in paradigm design affects the encoding of stimulus information.”